# CTLA-4 expression by B-1a B cells is essential for immune tolerance

Yang Yang [1✉], Xiao Li [2], Zhihai Ma[1], Chunlin Wang[3], Qunying Yang[3], Miranda Byrne-Steele[3], Rongjian Hong[4], Qing Min[4], Gao Zhou[2], Yong Cheng [5], Guang Qin[1], Justin V. Youngyunpipatkul[1], James B. Wing [6,7], Shimon Sakaguchi [7], Christian Toonstra [8], Lai-Xi Wang [8], Jose G. Vilches-Moure[9], Denong Wang [10], Michael P. Snyder [1], Ji-Yang Wang[4,11,12], Jian Han[3,13] & Leonore A. Herzenberg[1✉]

CTLA-4 is an important regulator of T-cell function. Here, we report that expression of this immune-regulator in mouse B-1a cells has a critical function in maintaining self-tolerance by regulating these early-developing B cells that express a repertoire enriched for auto-reactivity. Selective deletion of CTLA-4 from B cells results in mice that spontaneously develop autoantibodies, T follicular helper (Tfh) cells and germinal centers (GCs) in the spleen, and autoimmune pathology later in life. This impaired immune homeostasis results from B-1a cell dysfunction upon loss of CTLA-4. Therefore, CTLA-4-deficient B-1a cells up-regulate epigenetic and transcriptional activation programs and show increased self-replenishment. These activated cells further internalize surface IgM, differentiate into antigen-presenting cells and, when reconstituted in normal IgH-allotype congenic recipient mice, induce GCs and Tfh cells expressing a highly selected repertoire. These findings show that CTLA-4 regulation of B-1a cells is a crucial immune-regulatory mechanism.

[1] Department of Genetics, Stanford University School of Medicine, Stanford, CA, USA. [2] The Center for RNA Science and Therapeutics, Case Western Reserve University, Cleveland, OH, USA. [3] iRepertoire Inc, Huntsville, AL, USA. [4] Department of Immunology, School of Basic Medical Sciences, Fudan University, Shanghai, China. [5] St. Jude Children's Research Hospital, Memphis, TN, USA. [6] Laboratory of Human Immunology (Single Cell Immunology), World Premier International Immunology Frontier Research Center, Osaka University, Osaka, Japan. [7] Laboratory of Experimental Immunology, World Premier International Immunology Frontier Research Center, Osaka University, Osaka, Japan. [8] Department of Chemistry and Biochemistry, University of Maryland, College Park, MD, USA. [9] Department of Comparative Medicine, Stanford University School of Medicine, Stanford, CA, USA. [10] Tumor Glycomics Laboratory, SRI International Biosciences Division, Menlo Park, CA, USA. [11] Department of Clinical Immunology, Children's Hospital of Fudan University, Shanghai, China. [12] Department of Microbiology and Immunology, College of Basic Medical Sciences, Zhengzhou University, Zhengzhou, Henan, China. [13] HudsonAlpha Institute for Biotechnology, Huntsville, AL, USA. ✉email: yang71@stanford.edu; leeherz@stanford.edu

B cells participate in both the immune defenses and the maintenance of immune homeostasis. In the mouse, follicular B (FOB), marginal zone B (MZB), and CD5[+] B cells (B-1a) are the major mature B-cell subsets, which are distinguished from each other by the developmental origin, phenotype, anatomic location, and function[1–3]. B-1a cells are primarily generated during fetal development. As the earliest B cells that emerge during ontogeny, B-1a cells appear in the spleen much earlier than FOB and MZB cells. After their development ceases, B-1a cells are maintained via self-replenishment and persist in the adult animal, where they comprise a small B-cell subset in the spleen but are the dominant B-cell subset in the body cavities such as the peritoneal cavity[1–3].

Studies have demonstrated that B-1a and conventional B cells (e.g., FOB) arise from distinctive progenitor cells that appear at different times during ontogeny[4]. Earlier transfer studies show that B-cell progenitors in fetal liver efficiently support the B-1a generation, whereas the progenitors in adult bone marrow (BM) poorly generate B-1a cells despite their robust ability to support conventional B-cell generation[3,5,6]. Later studies further elucidate distinctive gene programs that operate in the fetal versus adult BM B-cell development pathways, i.e., Lin28b[+] Let-7[neg] B-cell precursors give rise to B-1a cells, whereas Lin28b[neg]Let-7[+] precursors support conventional B-cell development[7,8]. More recent studies have shown that B-1a cells are derived from the precursors around E8.5-E9.5 in the yolk sac and para-aortic splanchnopleura independent of the hematopoietic stem cells[9–11].

In accordance with these developmental distinctions, our previous study has demonstrate that the B-1a IgH repertoire differs dramatically from the repertories expressed FOB and MZB cells[12]. That is, after the first few weeks of de novo generation, the B-1a IgH repertoire undergoes potent V(D)J selection, leading to a repertoire that is less random and more repetitive than the repertoires expressed by FOB and MZB cells, and contains the highly selected V(D)J sequences that are shared by all adult animals. In addition, as animals age, B-1a IgH repertoire accumulates variants resulting from activation-induced cytidine deaminase (AID)-mediated hypermutation and class-switching[12]. Importantly, the B-1a repertoire V(D)J selection and diversification operate comparably under the germ-free condition, ruling out the microbiota-derived antigens as the driving force for these processes[12].

Our finding that the B-1a IgH repertoire selection processes proceed normally in germ-free mice support the notion that B-1a-cell generation reflects BCR-mediated positive selection by self-antigen(s)[12,13]. Hayakawa et al.[14] have cemented this concept by showing that the generation of the Thy-1-specific B-1a cells and their production of anti-Thy-1 natural antibodies require Thy-1 antigen. The dependence on BCR signaling for B-1a generation is further evident in studies showing that mutations that disrupt BCR signaling result in the decrease or even elimination of B-1a cells, whereas deletion of the negative regulators of BCR signaling increases their numbers[1,2].

The unique repertoire and the repertoire-defining mechanism underlie the antibody responses generated by B-1a cells. B-1a cells produce natural antibodies, the circulating antibodies that are secreted without stimulation by external antigens[2,15]. Intriguingly, many of these natural antibodies recognize moieties of self-antigens with low affinity and fulfill critical housekeeping functions such as clearance of cellular debris or metabolic waste[16–18]. As many of these conserved moieties are also expressed by pathogens, these pre-existing natural antibodies also provide the front line of the immune defense against the pathogen invasion[2,19–21].

The mechanisms that enable B-1a cells to produce a wide-range of functionally important autoreactive natural antibodies,

however, raise a question as to how the B-1a autoantibody production is regulated. This regulatory process must keep the risk of BCR-mediated self-antigen presentation and high-affinity auto-antibody-producing GC formation to a minimum. Studies presented here identify CTLA-4 expression by B-1a cells as a critical player in this control process, and demonstrate that when B-1a cells lose CTLA-4, they become dysfunctional and are able to induce Tfh cells and GC responses.

CTLA-4 is a key T-cell checkpoint regulator. Natural occurring Foxp3[+] regulatory T (Treg) cells constitutively express CTLA-4 and require it for their suppressive function[22,23]. After activation, non-Treg T cells also induce CTLA-4, which negatively regulates their proliferation and cytokine production[24,25]. The extreme potency of CTLA-4 in negatively regulating T-cell function has been well demonstrated in the fully CTLA-4-deficient ($Ctla4^{-/-}$) mice and CTLA-4 Treg knockout mice, all of which die early from fatal lymphoproliferative autoimmunity[22,26,27].

Here, we show the expression of CTLA-4 in B-1a cells, a B-cell subset that is derived from fetal development and express an Ig repertoire enriched for autoreactivity. Our studies also demonstrate the critical function of CTLA-4 in maintaining immune homeostasis by regulating these B cells.

## Results

**B-1a cells constitutively express CTLA-4.** CTLA-4 is commonly thought to be a T-cell-specific regulator. However, we report here that *Ctla4* s also expressed by B-1a cells in the spleen and PerC of adult mice (Fig. 1a). In contrast, it is not detectable in splenic FOB, MZB, and peritoneal B-2 cells (Fig. 1a). Our findings accord well with the released microarray data in Immunological Genome Project (ImmGen) database, which additionally shows that *Ctla4* is minimally expressed in B-cell progenitors, immature B cells in BM and spleen (Supplementary Fig. 1A). We further show that CD138[+] plasma cells (PCs), which are derived from B-1a cells and secret IgM in resting mice[15], also express *Ctla4* (Fig. 1a).

B-1a cells emerge during fetal development. They are readily detectable in neonatal spleen, whereas FOB and MZB cells have not yet been generated at this age (Supplementary Fig. 1B). By monitoring the splenic B-1a cells from neonatal life to adults, we found that B-1a *Ctla4* expression gradually increases during early ontogeny. Thus, it is detectable, albeit at very low levels, in splenic B-1a cells from day 5–7 neonates and gradually increases until adult life (at least 2 months) (Fig. 1b).

Intracellular fluorescence-activated cell sorting (FACS) analyses demonstrate that CTLA-4 is expressed by both splenic and peritoneal B-1a cells, albeit at levels that are lower than that expressed by Foxp3[+] Treg cells (Fig. 1c, d). Consistent with the *Ctla4* gene expression data, CTLA-4 expression level in splenic B-1a is lower than the level of their peritoneal counterpart (Fig. 1c, d). In contrast, CTLA-4 is not detected by splenic FOB, MZB and peritoneal B-2 cells (Fig. 1c, d). Constitutive CTLA-4 expression is also readily detected in splenic and peritoneal B-1a cells of T-cell-deficient ($Tcrb^{-/-}$ $Tcrd^{-/-}$) mice (Fig. 1c, d), excluding the idea that B-1a cells acquire CTLA-4 from the CTLA-4-expressing T cells.

CTLA-4 expression has been reported for human CD5[+] B chronic lymphocytic leukemia (B-CLL)[28,29]. Consistent with these findings, we detect an increased CTLA-4 expression by both splenic and peritoneal B-1a cells in Eμ-TCL1 transgenic mice (Fig. 1c, d), a mouse B-CLL model, in which B-1a cells develop into CLL-like tumors[30–32]. Importantly, despite the increased CTLA-4 expression in B-1a cells from Eμ-TCL1 mice, CTLA-4 is still undetectable in FOB and peritoneal B-2 cells of these mice (Fig. 1c, d). Collectively, these findings demonstrate

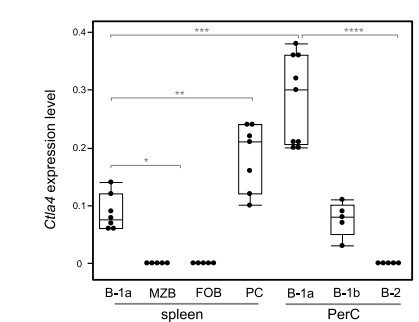

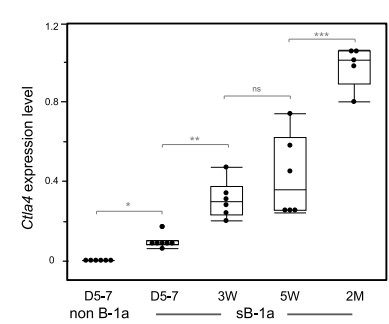

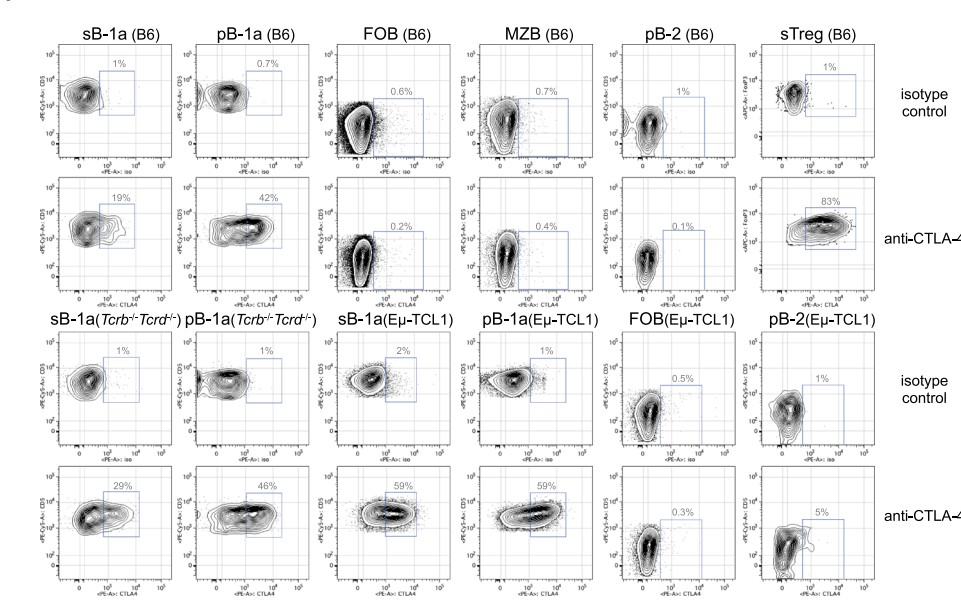

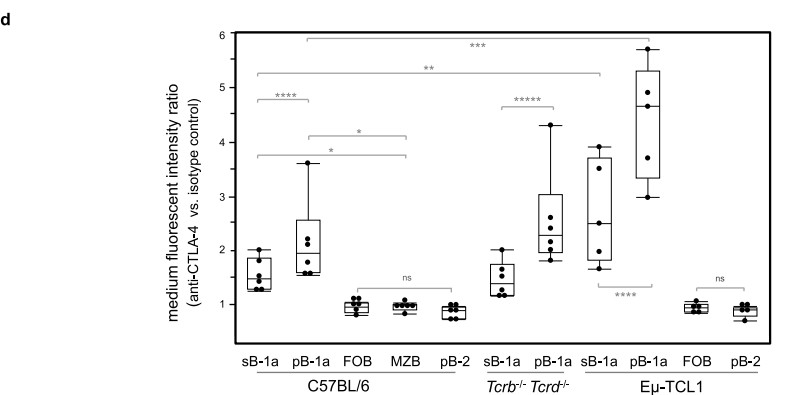

that, like CD5 expression, CTLA-4 expression selectively marks B-1a cells within the resting B-cell compartment.

CTLA-4 binds its B7 ligands (CD80 and CD86) with high affinity and avidity[33]. These ligands are largely expressed on antigen-presenting cells (APCs)[33]. We show that both splenic and peritoneal B-1a cells express CD80 and CD86 (Supplementary Fig. 1D, E). Notably, CD80, which has the strongest affinity with CTLA-4[33], is expressed at a much higher level on B-1a cells than the levels expressed on FOB, MZB, and splenic CD11c[+] dendritic cells (DCs) (Supplementary Fig. 1D, E). Therefore, although B-1a cells express low levels of CTLA-4, their co-expression of the high levels of B7 ligands, particular

CD80, suggest that CTLA-4 expressed by B-1a cells may uniquely have greater access to its ligands.

**B-1a cells have increased activation in CTLA-4 B-cell CKO mice.** To study the function of CTLA-4 on B-1a cells, we selectively deplete CTLA-4 from B cells by generating the CTLA-4 B-cell conditional knockout (CKO) mice (Ctla4[fl/fl] Cd19[Cre/+])[22,34]. In these CKO mice, T-cell subsets such as Treg cells express the floxed Ctla4 genotype, whereas B-1a cells have already lost Ctla4 owing to Cre-mediated recombination (Supplementary Fig. 2A). As a result, CTLA-4 expression is lost in B-1a cells but remains normal in Treg cells of these animals (Supplementary Fig. 2B, C).

**Fig. 1 CTLA-4 is selectively expressed by B-1 cells within the resting B-cell compartment. a** *Ctla4* expression by mature B-cell subsets in adult C57BL/6J mice (2–3 months old) was measured by qRT-PCR. B-cell subsets were phenotypically defined as: B-1a, CD19$^+$ IgM$^{hi}$ IgD$^{lo/-}$ CD21$^{lo/-}$ CD43$^+$ CD5$^+$; B-1b, CD19$^+$ IgM$^{hi}$ IgD$^{lo/-}$ CD43$^+$ CD5$^{neg}$; MZB, CD19$^+$ IgM$^{hi}$ IgD$^{lo/-}$ CD21$^{hi}$ CD43$^{neg}$ CD5$^{neg}$; FOB and peritoneal B-2, CD19$^+$ IgM$^{lo}$ IgD$^{hi}$ CD43$^{neg}$ CD5$^{neg}$; PCs, CD19$^+$ IgD$^{neg}$ CD138$^+$ CD267$^+$. *Ctla4* expression levels are shown as the data relative to the *Ctla4* level expressed by splenic CD3$^+$ CD4$^+$ CD25$^+$ Treg cells (>90% Foxp3$^+$) using comparative C$_T$ method $2^{-\Delta\Delta C_T}$. Each dot represents data for an individual mouse, $n = 5$–9 mice per subset. *$p < 0.003$, **$p < 0.007$, ***$p < 0.001$, ****$p < 0.002$. **b** *Ctla4* expression by splenic B-1a (sB-1a) and non B-1a cells in neonatal, young or adult mice was measured by qRT-PCR. D, day, W, week, M, month. FACS gating is shown in Supplementary Fig. 1B. *Ctla4* expression levels are shown as the data relative to the *Ctla4* level expressed by adult sB-1a. $n = 5$–7 mice per group, *$p < 0.001$, **$p < 0.003$, ***$p < 0.006$, ns, not significant ($p < 0.5$). **c** CTLA-4 expression by sB-1a, peritoneal B-1a (pB-1a), FOB, MZB, peritoneal B-2 (pB-2), and splenic Treg (sTeg) cells from indicated mice were measured by intracellular CTLA-4 FACS analysis. FACS gating is shown in Supplementary Fig. 1C. *y* axis shows surface CD5 expression for B-cell subsets and intracellular Foxp3 expression for sTreg cells, *x* axis shows data for cells stained with phycoerythrin (PE)-conjugated anti-CTLA-4 or isotype control antibodies. **d** Data summarizing independent FACS analyses ($n = 6$) is shown, each dot represents data for an individual mouse. *y* axis shows ratio of medium fluorescent intensity (MFI) values of the indicated B-cell subsets stained with PE-conjugated anti-CTLA-4 vs. isotype control antibody. *$p < 0.0001$, **$p < 0.03$, ***$p < 0.01$, ****$p < 0.05$, *****$p < 0.007$, ns, not significant ($p < 0.6$). Box plots in **a**, **b**, **d**: box draws 75% (upper), 50% (center line), and 25% (down) quartile, the maxima and minima outliers are shown as top and bottom line, respectively. Statistical significance was tested using nonparametric Wilcoxon one-way test.

Bromodeoxyuridine (BrdU)-incorporation studies demonstrate that B-1a cell self-replenishment in CKO mice is increased. Both splenic and peritoneal B-1a cells in CKO mice incorporate more BrdU (BrdU$^+$) than their control counterparts (*Ctla4$^{+/+}$ Cd19$^{Cre/+}$*) (Fig. 2a). As a result, the B-1a compartment is expanded in CKO mice (Supplementary Fig. 2D). Notably, in each individual animal, splenic B-1a cells always contain a much higher proportion of BrdU$^+$ cells than peritoneal B-1a cells (Fig. 2a), supporting the idea that spleen is the principal site where B-1a cells undergo self-replenishment[12,35,36].

CTLA-4-deficient B-1a cells in CKO mice also upregulate surface expression of the major histocompatibility class II (MHC II) and the signaling lymphocytic activation molecule (SLAM), a marker expressed at a high level in GC B cells (Supplementary Fig. 2E). Consistent with this increased activation, the B-1a IgH repertoire of CKO mice contains higher levels of somatic hypermutation (SHM) in IgV$_H$ than their control counterparts (Fig. 2b).

The increased B-1a turnover and activation in CKO mice is owing to that CTLA-4-deficient B-1a cells upregulate epigenetic and transcriptional activation programs. RNA-seq analysis show that among 10$^4$ detected transcripts, ~130 genes are differentially expressed by B-1a cells from CKO mice vs. B-1a cells from controls. About 90% of these genes are upregulated in CTLA-4-deficient B-1a cells. Strikingly, the most significantly upregulated transcripts are enriched for genes participating in chromatin methylation and acetylation including *Tet3*, which catalyzes the DNA cytosine demethylation; *Kmt2d*, *Kmt2b*, and *Setd1b*, which catalyze histone H3 Lys-4 methylation; and, *Ep300*, *Ep400*, *Kat6a*, and *Brd4*, which catalyze histone acetylation (Fig. 2c, d, Supplementary Fig. 2F), all being the epigenetic tags for transcriptional activation.

Coinciding with this epigenetic activation, genes of transcription activator, PI3K/Jak-Stat BCR signaling pathways, and cell cycle pathways are upregulated in CTLA-4-deficient B-1a cells (Fig. 2c, d). In particular, cell cycle component Cyclin D2 gene, i.e., *Ccnd*2, which has been shown to be essential for B-1a generation[37], is substantially elevated (Fig. 2d). Based on these findings, we conclude that CTLA-4 functions to constrain B-1a cell activation and this negative regulation is mediated by epigenetic and transcriptional reprogramming.

**Tfh cells and GCs spontaneously arise in spleens of CTLA-4 CKO mice.** Concurrent with the B-1a cell activation in CKO mice, GC responses spontaneously arise in the spleen. Thus, IgM$^{neg}$ IgD$^{neg}$ B cells are drastically increased (Fig. 3a, Supplementary Fig. 3A) and over 60% of these cells express hallmarks of GC B cells, i.e., they proliferate (Ki67$^+$), turn off CD38, and

upregulate CD95, PNA-glycan, GL7, Bcl6, and AID (Supplementary Fig. 3B, C). Further, a substantial proportion of IgM$^{neg}$ IgD$^{neg}$ B cells express surface IgG1 and hence have undergone class-switching (Fig. 3a). IgH sequencing of GC B cells (CD95$^+$ CD38$^{neg}$) confirms high levels of SHM. About 40% of IgH transcripts contain more than five substituted nucleotides within the V$_H$ region spanning from the complementarity determining region 2 (CDR2) to CDR3. This translates to a SHM rate of ~25 per 10$^3$ base pair (Fig. 2b).

Depleting CD4 T cells or blocking CD40 signaling prevents GC formation in CKO mice (Fig. 3b), indicating that akin to the GC responses induced by foreign antigens, the spontaneous GC responses depend on CD4 T-cell and CD40 signaling. Consistent with this conclusion, CD4 Tfh cells (PD-1$^{hi}$ CXCR5$^+$ ICOS$^+$ Foxp3$^{neg}$ Bcl6$^+$), the T effector cells that play crucial roles in the T-dependent GC responses[38–41], spontaneously arise in spleens of CKO mice (Fig. 3c, d).

Notably, Tfh cells that arise in the spleen of CKO mice are barely detectable in the lymph node (LN) of these mice (Supplementary Fig. 3D). Consistently, LN contains fewer GC B cells than spleen (Supplementary Fig. 3E). These findings indicate that spleen is the main lymphoid organ that supports the generation of the Tfh cells and GC responses in CKO mice.

**CTLA-4 CKO mice generate autoantibodies and late-onset autoimmunity.** GC responses in CKO mice are detectable in spleens ~6 weeks after birth and are sustained thereafter (Supplementary Fig. 4A). As a result, numbers of class-switched B cells and the levels of serum IgG and IgE rise progressively as animals age (Supplementary Fig. 3A, Fig. 4a). Importantly, sera from CKO mice contain IgG and IgE autoantibodies reactive with double-strand DNA (dsDNA), nuclear antigen, and rheumatoid factor (RF) (Fig. 4b, c, Supplementary Fig. 4B). These antibodies increase with age but are minimally detectable in age-matched control mice (Fig. 4b, c, Supplementary Fig. 4B).

Furthermore, similar to the normal mice, sera from control mice rarely contain IgG anti-glycan autoantibodies (Fig. 4d). In contrast, sera from CKO mice have autoreactive IgG reacting with *N*-glycans, e.g., oligo-mannoses, asialo- and aglacto-glycans, and *O*-glycan core-based autoantigens (Fig. 4d). Expressed on the surface of apoptotic cells, these glycan moieties may serve as targets for natural antibodies. Consistent with this notion, both control and CKO mouse sera contain natural IgM that reacts with these glycan autoantigens, such as Man1Gn2Asn, Man9Gn2Asn, Glc1Man9, and GlcNAc-RB (Fig. 4d). In essence, although sera of control and CKO mice show similar natural IgM anti-glycan profiles, their IgG profiles differ in that only the sera from CKO mice contain autoreactive IgG to these glycans.

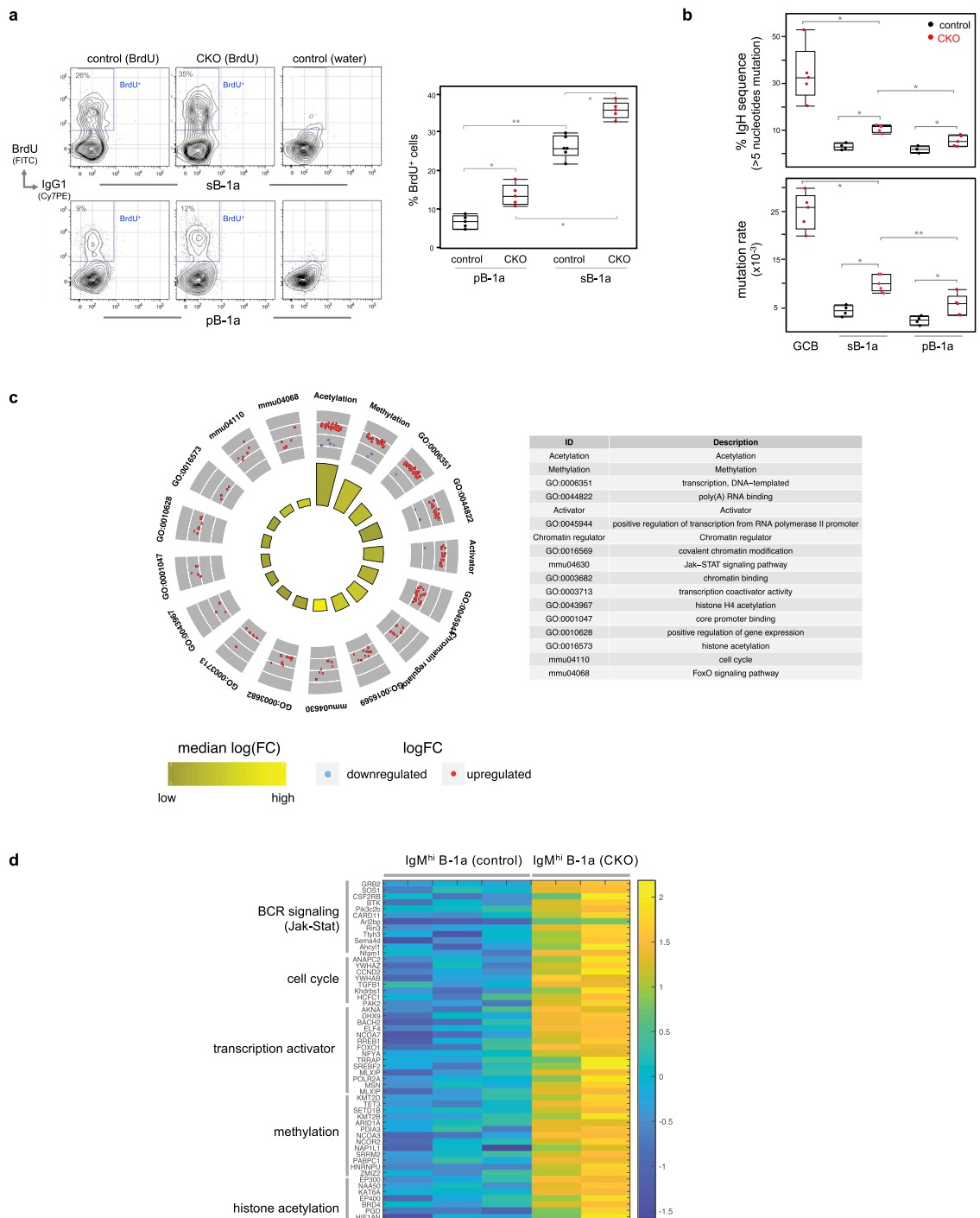

**Fig. 2 B-1a cells in CKO mice show increased self-replenishment and upregulate activation gene programs. a** Control or CKO mice were fed with BrdU-containing water for 6 days and BrdU incorporation (BrdU$^+$) by splenic and peritoneal B-1a cells in these mice was measured by FACS analysis. B-1a cells were gated as CD19$^+$ IgM$^{hi}$ IgD$^{lo/neg}$ CD21$^{lo/neg}$ CD43$^+$ CD5$^+$. Data summarizing the percentages of BrdU$^+$ cells in sB-1a or pB-1a cells for five independent experiments is shown. Each dot represents data for an individual mouse. *$p < 0.01$, **$p < 0.006$, nonparametric Wilcoxon one-way test. **b** sB-1a, pB-1a, and GC B (CD19$^+$ CD95$^+$ CD38$^{neg}$) cells in control or CKO mice were sorted and then sequenced to obtain their IgH transcripts. The percentages of IgH transcripts containing >5 nucleotide changes and the mutation rate for each population are shown, $n = 4$–5 biologically independent samples per subset, each sample was from an individual mouse, *$p < 0.01$, **$p < 0.02$, nonparametric Wilcoxon one-way test. **c** Circular visualization graph shows the selected GO terms for differentially expressed genes between splenic B-1a cells in control and CKO mice. Each dot in the outer circle represents an individual upregulated (red) or downregulated (blue) gene. Genes of the same functional term are grouped together and the detail of each group is shown in the right table. The colored bars in the inner circle summarize the median folds change of the genes sharing the same functional term. **d** Heatmap shows the expression levels of listed genes measured by RNA-seq. Each row represents data of an individual gene. Genes with the same functional terms are grouped together. Each column represents data for an individual splenic IgM$^{hi}$ B-1a cell sample from control or CKO mice. All mice are 2–3 months old. Box plots in **a**, **b**: box draws 75% (upper), 50% (center line) and 25% (down) quartile, the maxima and minima outliers are shown as top and bottom line, respectively.

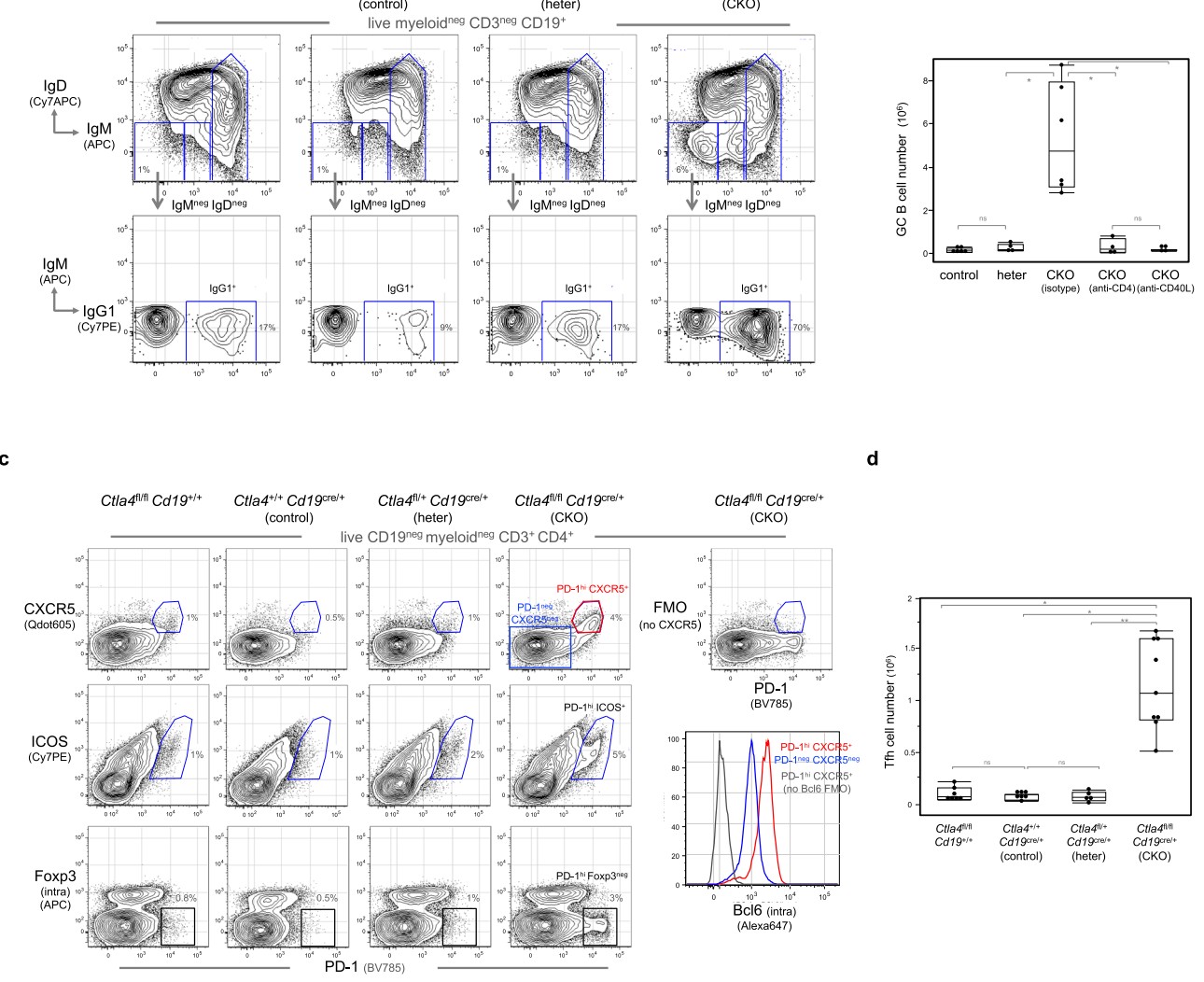

**Fig. 3 T-dependent GC responses and Tfh cells arise spontaneously in spleens of CKO mice. a** Spleen cells from indicated mice were analyzed by FACS. Live myeloid$^{neg}$ CD3$^{neg}$ CD19$^+$ splenic B cells were gated to show IgM$^{neg}$ IgD$^{neg}$ B cells, which were further gated to reveal IgG1$^+$ cells. **b** Data summarizing the numbers of GC B cells (CD19$^+$ CD38$^{neg}$ CD95$^+$) in spleens of control, heterozygous and CKO mice treated with indicated antibodies is shown. Each dot represents data for an individual mouse, $n = 4–6$ mice per mouse group. *$p < 0.01$, ns, not significant ($p < 0.1$), nonparametric Wilcoxon one-way test. **c** Live myeloid$^{neg}$ CD19$^{neg}$ CD3$^+$ CD4$^+$ splenic T cells from indicated mice were gated to show PD-1, CXCR5, ICOS, and intracellular Foxp3 expression. Fluorescent minus one (FMO) staining, in which the fluorescent anti-CXCR5 antibody was omitted from the staining cocktail, is used to define CXCR5-expressing cells. FACS histogram on the right shows the intracellular Bcl6 expression by Tfh (PD-1$^{hi}$ CXCR5$^+$, red line) and non-Tfh cells (PD-1$^{neg}$ CXCR5$^{neg}$, blue line) in CKO mice. **d** Data summarizing numbers of Tfh cells (CD3$^+$ CD4$^+$ PD-1$^{hi}$ CXCR5$^+$) in spleens of each mouse group is shown, $n = 5–9$ mice per mouse group, *$p < 0.001$, **$p < 0.003$, ns, not significant ($p < 0.1$), All mice are 2–3 months old. Box plots in **c**, **b**: box draws 75% (upper), 50% (center line), and 25% (down) quartile, the maxima and minima outliers are shown as top and bottom line, respectively.

Differing from fully CTLA-4-deficient ($Ctla4^{-/-}$) and CTLA-4 Treg knockout mice, which all die at an earlier age, CTLA-4 B-cell CKO mice live much longer (over 1 year) and do not manifest notable pathologies until they are older than 7 months of age, e.g., some of these older CKO mice develop pruritic and other immune pathologies (Supplementary Fig. 4C). Microscopic analysis of three such mice reveals skin pathology that includes accumulation of dense eosinophilic matrix (sometimes resembling amyloid; Supplementary Fig. 4D). In addition, some mice have varying degrees of dermal and intra-epidermal clefting and ulceration, a feature shared with some human autoimmune disorders (e.g., pemphigus family) (Supplementary Fig. 4D). Finally, some of these mice have inflammatory infiltration (neutrophilic and/or lymphocytic) in various organs including

the small intestine (2/3), ears (1/3), and liver (1/3), (Supplementary Fig. 4E).

**Activated CTLA-4-deficient B-1a cells differentiate into APCs.** Like IgM$^{neg}$ IgD$^{neg}$ B cells, IgM$^{int}$ IgD$^{neg}$ B cells also increase in spleens of CTLA-4 B-cell CKO mice. Surprisingly, both populations contain a B-1a subset (CD5$^+$ CD43$^+$ B220$^{lo}$) (Fig. 5a). Thus, differing from the typical B-1a cells that express high levels of surface IgM (IgM$^{hi}$), these two B-1a subsets detected in CKO mice express intermediate (IgM$^{int}$) to undetectable surface IgM (IgM$^{neg}$). Importantly, in spite of being surface IgM negative, IgM$^{neg}$ B-1a cells express the intracellular IgM and the expression levels are much lower than the intracellular levels of IgM-secreting cell (Fig. 5a). This suggests that the intracellular IgM

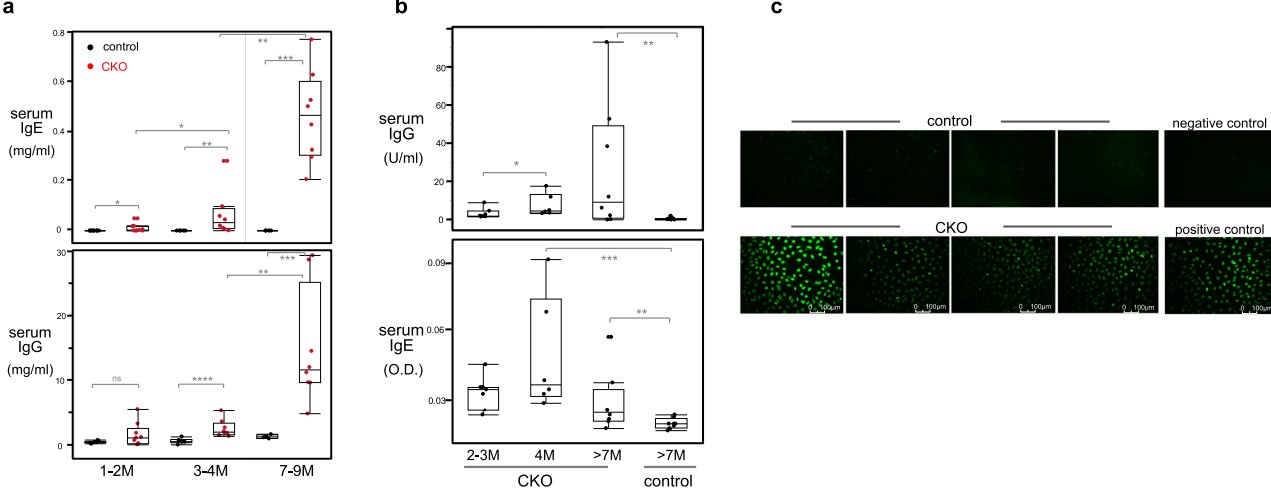

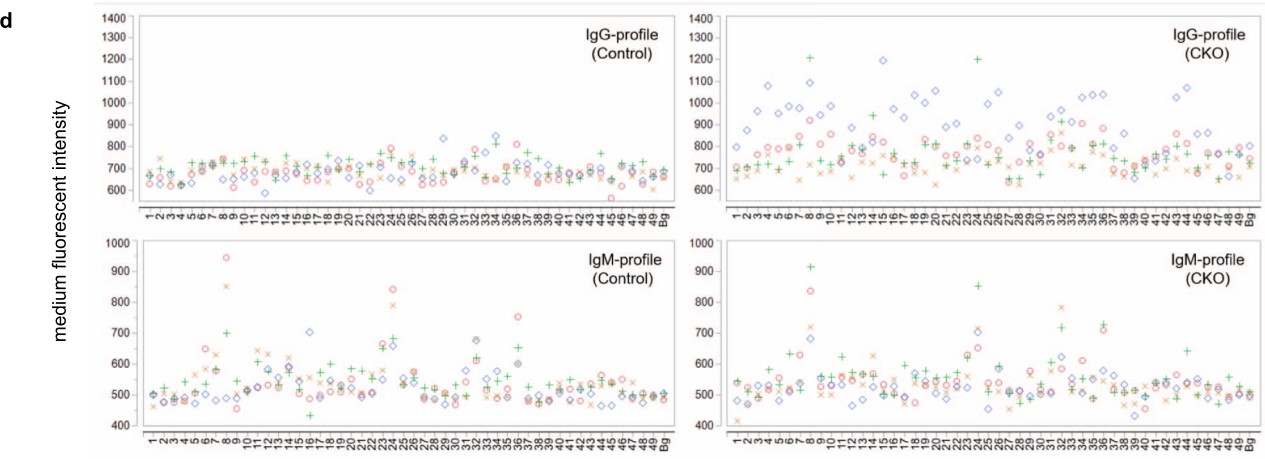

carbohydrate autoantigens ID

**Fig. 4 CKO mouse serum contains autoantibodies to dsDNA, nuclear antigen, and glycan autoantigens. a** IgG and IgE levels in serum of age-indicated control or CKO mice were measured by Elisa, $n = 6$–8 mice per mouse group, $*p < 0.03$, $**p < 0.001$, $***p < 0.006$, $****p < 0.002$, ns, not significant ($p < 0.2$), nonparametric Wilcoxon one-way test. **b** Anti-dsDNA IgG and IgE levels in serum of age-indicated CKO and control mice were measured by Elisa, $n = 5$–8 mice per mouse group, $*p < 0.05$, $**p < 0.03$, $***p < 0.02$, nonparametric Wilcoxon one-way test. **c** Anti-nuclear antigen (ANA) autoantibody in serum of CKO or control mice (7–14 months old) was tested, $n = 8$ mice per mouse group. Representative data for four mice from each group are shown. None (0/8) of the control mice and 5/8 of CKO mice show positive for ANA. **d** Anti-glycan IgG and IgM autoantibodies in serum of control or CKO mice were measured by carbohydrate microarray assay. Overlay plots of anti-glycan autoantibody IgG (upper) and IgM (lower) profiles of serum from four control or CKO mice (>7-month-old) are shown. Each symbol represents data of IgG or IgM reacting to an individual glycan antigen for an individual mouse serum. *y*-axis: the mean values of triplicate microspot detections of individual sera sample for each antigen, *x*-axis: corresponding ID of glycan antigens. Oligo-mannoses (#8–34), asialo- and aglacto-glycans(#3–6), *O*-glycan core-based autoantigens (#43–44), Man1Gn2Asn (#8), Man9Gn2Asn (#24), Glc1Man9 (#32), GlcNAc-RB (#36). Information of each glycan antigen is shown in Supplementary Table 6. The mean values of background microspots (Bg) of each staining are shown in the rightmost of *X* axis in each graph. Box plots in **a**, **b**: box draws 75% (upper), 50% (center line), and 25% (down) quartile, the maxima and minima outliers are shown as top and bottom line, respectively.

detected in the IgM$^{neg}$ B-1a cells are derived from the internalized surface IgM, rather than from the secreted intracellular IgM.

Importantly, over 15% of these IgM$^{neg}$ B-1a cells express a differentiated phenotype, i.e., CD95$^{+}$ CD38$^{lo}$ GL7$^{+}$ CD150$^{+}$ (Fig. 5b, c). In contrast, these differentiated cells are barely detected in control mice (Supplementary Fig. 5A, Fig. 5c). Coinciding with this differentiation, the CD95$^{+}$ CD38$^{lo}$ GL7$^{+}$ CD150$^{+}$ IgM$^{neg}$ B-1a cells express class-switched Ig on surface (csIg$^{+}$) and downregulate CD5 (Supplementary Fig. 5B). Strikingly, this differentiated phenotype (CD95$^{+}$ CD38$^{lo}$ GL7$^{+}$ CD150$^{+}$ csIg$^{+}$) is also expressed by IgM$^{neg}$ IgD$^{neg}$ B cells in

the thymus (Supplementary Fig. 5C), the thymic B cells that have recently been reported to be efficient APCs that present self-antigens to developing thymocytes[42].

We further found that most IgM$^{neg}$ IgD$^{neg}$ thymic B cells are CD5$^{+}$ CD43$^{+}$ (Supplementary Fig. 5D), supporting previous reports that many thymic B cells are B-1a cells[43–45]. Our finding that CD95$^{+}$ CD38$^{lo}$ GL7$^{+}$ CD150$^{+}$ IgM$^{neg}$ B-1a in CKO mice express the same phenotype as the thymic IgM$^{neg}$ IgD$^{neg}$ B-1a cells, the APCs in the thymus, suggests that these differentiated IgM$^{neg}$ B-1a cells in CKO mice are APCs. Consistent with this idea, these cells express increased surface MHC II and

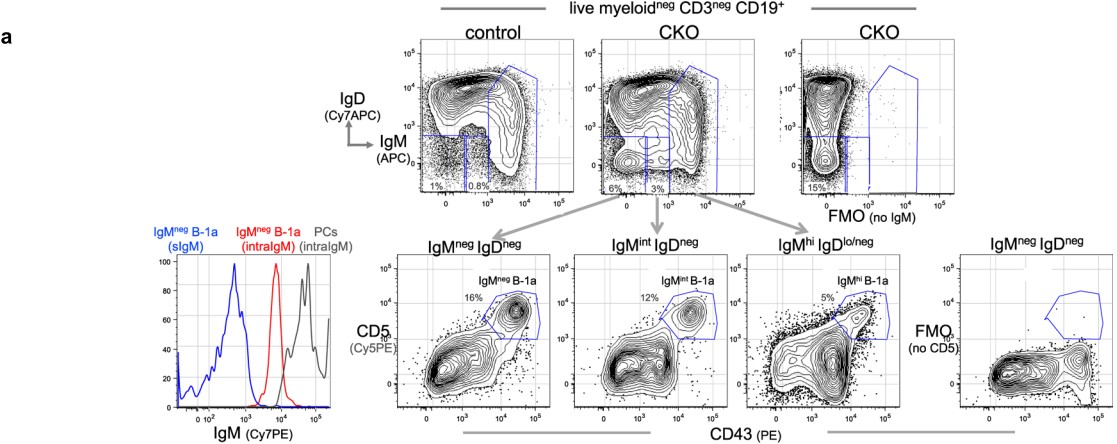

intracellular H2-DM (Supplementary Fig. 5E), a chaperone that has a central role in antigenic loading of the MHC II molecule[46].

The CD95+ CD38lo GL7+ PNA+ CD150+ IgMneg B-1a cells found in spleen of CKO mice, however, have not yet fully differentiated to GC B cells (CD95+ CD38neg GL7hi PNAhi CD150+), which further downregulate CD38 and upregulate GL7 and PNA (Supplementary Fig. 5F). We call CD95+ CD38lo GL7+ PNA+ CD150+ IgMneg B-1a cells pre-GC APCs, the APCs prior to GC formation, to distinguish them from the APCs in the fully developed GC. Nevertheless, they express comparable levels of intracellular H2-DM and its inhibitor, H2-DO, as CXCR4lo CD86hi light zone GC B cells (Supplementary Fig. 5G), the

**Fig. 5 IgM$^{hi}$ B-1a cells in CKO mice internalize surface IgM and start expressing the pre-GC APC phenotype. a** Live myeloid$^{neg}$ CD3$^{neg}$ CD19$^{+}$ splenic B cells in control or CKO mice were gated to show IgM and IgD expression. IgM$^{hi}$ IgD$^{lo/neg}$, IgM$^{int}$ IgD$^{neg}$, and IgM$^{neg}$ IgD$^{neg}$ B cells in CKO mice were gated to reveal IgM$^{hi}$, IgM$^{int}$, and IgM$^{neg}$ B-1a cells, based on CD5 and CD43 expression. Two FMO stainings were used to define IgM and CD5 positive cells. FACS histogram on the left shows the intracellular IgM level in IgM$^{neg}$ B-1a cells (red line) and IgM-secreting PCs (IgD$^{neg}$ CD138$^{+}$) (gray line). Blue line shows the surface IgM staining level in IgM$^{neg}$ B-1a cells that were stained with anti-IgM antibody before fixation and permibilization. **b** FACS plots shows cells expressing the differentiated CD95$^{+}$ CD38$^{lo}$ GL7$^{+}$ CD150$^{+}$ phenotype in IgM$^{hi}$, IgM$^{int}$, and IgM$^{neg}$ B-1a cells from CKO mice. **c** Data summarizing the percentage of CD95$^{+}$ CD38$^{lo}$ GL7$^{+}$ CD150$^{+}$ cells in IgM$^{hi}$, IgM$^{int}$ and IgM$^{neg}$ B-1a cells from control and CKO mice is shown. Each dot represents data for an individual mouse, $n = 7$–10 mice per subset, *$p < 0.001$, **$p < 0.002$, ns, not significant ($p < 0.2$), nonparametric Wilcoxon one-way test. **d** IgM$^{hi}$ B-1a and IgM$^{neg}$ B-1a cells from CKO mouse were sorted and then sequenced to obtain their IgH transcripts. Data summarizing the frequencies of IgH sequences containing >5 nucleotide changes and the mutation rate of each sample is shown, $n = 7$ samples per group with each sample from an individual mouse. Data for IgM$^{hi}$ B-1a and IgM$^{neg}$ B-1a cells from the same mouse are connected. *$p < 0.001$, nonparametric Wilcoxon one-way test. All mice are 2 months old. Box plots in **c**, **d**: box draws 75% (upper), 50% (center line), and 25% (down) quartile, the maxima and minima outliers are shown as top and bottom line, respectively.

**Table 1 IgH repertoire expressed by IgM$^{neg}$ B-1a cells in CKO mice contains class-switched IgH transcripts.**

| CTLA-4 B-cell CKO | B-1a subsets | % IgH isotype transcripts | | | | | | | |
|---|---|---|---|---|---|---|---|---|---|
| | | IgM | IgD | IgG3 | IgG1 | IgG2b | IgG2c | IgE | IgA |
| #1 | IgM$^{hi}$ | 97.9 | | | | | | 0.7 | 1 |
| | IgM$^{neg}$ | 32.2 | | 3.2 | 15.5 | 8.9 | 2.6 | 32.7 | 4.9 |
| #2 | IgM$^{hi}$ | 99.9 | | | | | | | |
| | IgM$^{neg}$ | 33.2 | | 31.1 | 4.9 | 17.3 | 8.6 | 1.7 | 3.1 |
| #3 | IgM$^{hi}$ | 99.3 | | 0.5 | | | | | |
| | IgM$^{neg}$ | 30.6 | | 9.1 | 20.2 | 12.3 | 4.6 | 18.4 | 4.8 |
| #4 | IgM$^{hi}$ | 99.3 | | 0.1 | | | | | |
| | IgM$^{neg}$ | 65.4 | 4.1 | 4.4 | 6.4 | 2.7 | 1.4 | 4.8 | 10.8 |
| #5 | IgM$^{hi}$ | 99.4 | 0.5 | | | | | | 0.1 |
| | IgM$^{neg}$ | 84.7 | 0.4 | 1 | 2 | 0.8 | 0.5 | 5.3 | 5.3 |
| #6 | IgM$^{hi}$ | 98 | 0.5 | | | | | 0.7 | |
| | IgM$^{neg}$ | 6.4 | 0.1 | 2.2 | 5 | 0.9 | 0.9 | 15 | 69.5 |
| #7 | IgM$^{hi}$ | 99.2 | | | | | | | |
| | IgM$^{neg}$ | 45.9 | 0.1 | 0.7 | 27.5 | 15.4 | 1.8 | 4.5 | 4.1 |

IgM$^{hi}$ B-1a and IgM$^{neg}$ B-1a cells of CTLA-4 B-cell CKO mice were sorted and sequenced to obtain their IgH transcripts in each population. Percentage of IgH transcript expressing indicated Ig isotype for IgM$^{hi}$ B-1a and IgM$^{neg}$ B-1a cells from the indicated CKO mouse is shown in the column, $n = 7$ mice (2–3 months old).

APCs that undergo antigen-selection by presenting antigens to T cells in the GC.

In contrast, when compared with the IgM$^{neg}$ B-1a cells, the IgM$^{hi}$ B-1a cells in the same CKO mice contain many fewer CD95$^{+}$ CD38$^{lo}$ GL7$^{+}$ CD150$^{+}$ cells (Fig. 5b, c) and do not express csIg$^{+}$ on surface (Supplementary Fig. 5B). Consistent with these findings, IgH sequencing of IgM$^{neg}$ and IgM$^{hi}$ B-1a cells from the same CKO mice show that class-switched IgG, IgA, and IgE transcripts are readily detected in the IgM$^{neg}$ B-1a repertoire, whereas IgM is the dominant Ig (>98%) for IgM$^{hi}$ B-1a cells (Table 1). In addition, the IgM$^{neg}$ B-1a cells not only show class-switching, but also have higher levels of IgV$_H$ SHM than their IgM$^{hi}$ B-1a counterparts (Fig. 5d).

Importantly, IgM$^{neg}$ and IgM$^{hi}$ B-1a cells share more than the CD5$^{+}$ CD43$^{+}$ B220$^{lo}$ phenotype. Within a given CKO animal, they show extensive IgH repertoire-sharing, i.e., both repertoires contain an array of identical V(D)J sequences that we have previously showed to be positively selected and highly conserved in C57BL/6 B-1a IgH repertoires[12]. These include the VH11 encoded IgH specific for phosphatidyl choline (PtC) and T15 idiotype IgH specific for phosphoryl choline (PC) (Supplementary Table 1). Most importantly, for these prototypic B-1a V(D)J sequences, IgM is always the dominant (~99%) isotype in IgM$^{hi}$ B-1a, whereas class-switched IgG, IgA, and IgE are readily detectable among IgM$^{neg}$ B-1a cells in the same CKO mice (Supplementary Table 2). Some of these class-switched Ig transcripts from IgM$^{neg}$ B-1a cells already show SHM (Supplementary Fig. 5H).

Taken together, we conclude that IgM$^{neg}$ B-1a cells in CKO mice represent further differentiated descendants of IgM$^{hi}$ B-1a cells. In essence, the activated CTLA-4-deficient B-1a cells in CKO mice undergo further differentiation, i.e., they internalize surface IgM and start to express phenotype of the pre-GC APCs (CD95$^{+}$ CD38$^{lo}$ GL7$^{+}$ PNA$^{+}$ CD150$^{+}$ csIg$^{+}$).

**IgM$^{neg}$ B-1a cells in CKO mice express APC gene programs.** Along with the internalization of surface BCR and the phenotypic change to APCs, IgM$^{neg}$ B-1a cells in CKO mice downregulate transcriptional activation programs but concomitantly turn on gene programs, that indicates their functional shift from proliferation-promoting transcriptional activation towards antigen processing and presentation. RNA-seq analyses show that ~1400 genes are differentially expressed between IgM$^{hi}$ B-1a and IgM$^{neg}$ B-1a cells in CKO mice. About 60% of these differentially expressed genes are downregulated in IgM$^{neg}$ B-1a cells compared with IgM$^{hi}$ B-1a cells, including genes participating in epigenetic and transcriptional activation, cell cycle progression, and BCR signaling (Supplementary Fig. 6). Therefore, in sharp contrast to the CKO IgM$^{hi}$ B-1a cells, which upregulate epigenetic reprogramming and transcriptional activation programs (Fig. 2c, d), the IgM$^{neg}$ B-1a cells in CKO mice turn off these programs (Fig. 6a).

In parallel, however, these IgM$^{neg}$ B-1a cells in CKO mice turn on gene programs that are associated with antigen processing and presentation. Thus, genes associated with antigen trafficking and processing are upregulated in IgM$^{neg}$ B-1a cells, such as those

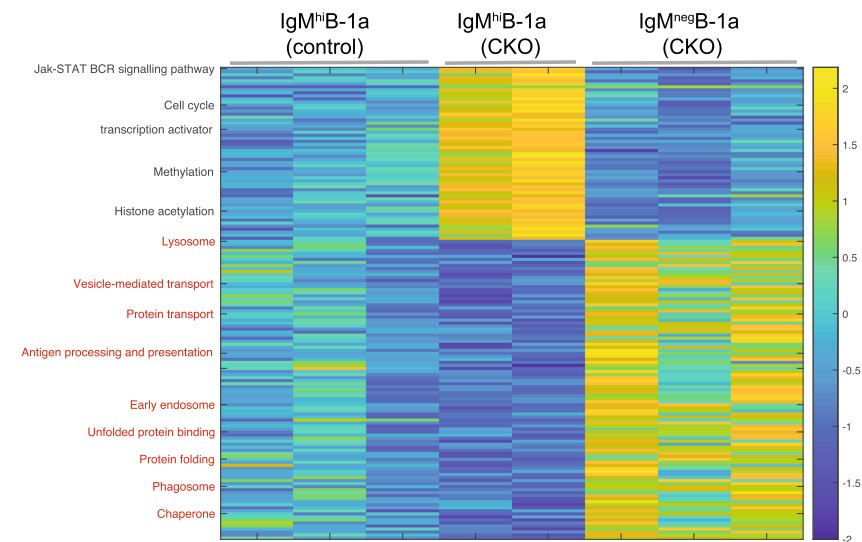

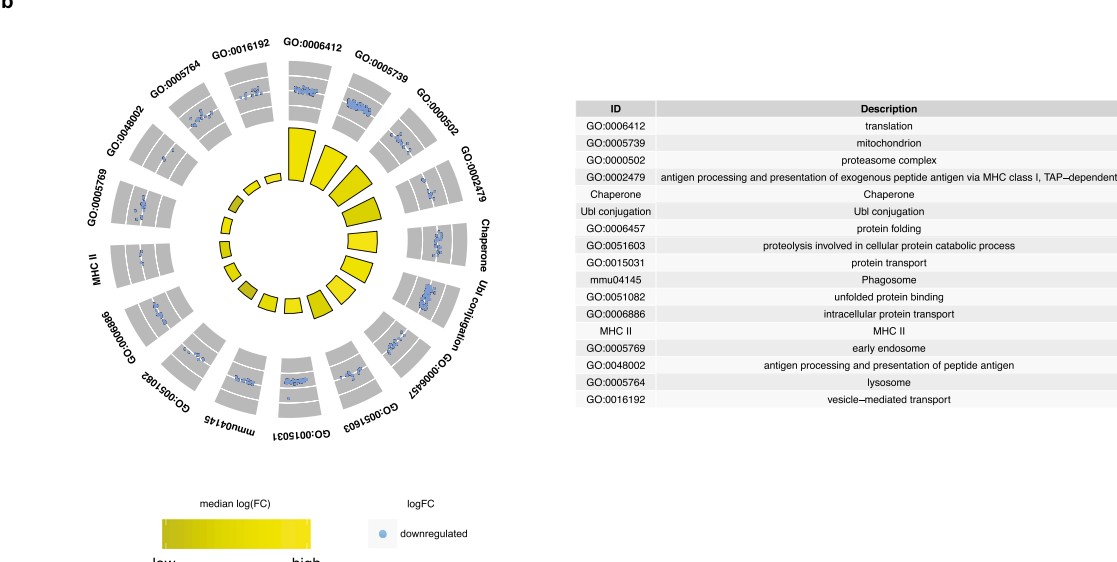

**Fig. 6 IgM^neg B-1a in CKO mice express a highly coordinated gene program indicating a function shift to APCs. a** Heat-map shows the expression levels of the selected genes measured by RNA-seq. Row-rise standardization was performed for visualization purpose. Each row represents data of an individual gene. Genes of the same functional terms are grouped together. Each column represents the data of each indicated B-1a sample. **b** Circular visualization graph shows the selected GO terms for differentially expressed genes between sB-1a from control and CKO mice. Each dot in the outer circle represents an individual upregulated gene. Genes of the same functional term are grouped together and the detail of each group is shown at the right table. In the inner circle, the colored bars summarize the median fold change of the genes sharing the same functional term.

involved in ubiquitin and protease degradation, protein folding in the endoplasmic reticulum, and vesicle-mediated transport within earlier endosome, lysosome, and Golgi (Fig. 6a, b). Accordingly, genes involved in antigen presentation via the MHC pathway are also upregulated in IgM^neg B-1a cells, including the gene of H2-DM, a key player in the removal of the CLIP peptide from MHC II, which allows loading with antigenic peptides (Fig. 6a, b).

Taking the preceding findings together, we conclude that IgM^neg B-1a cells in CKO mice express a highly coordinated gene program that operates as their transcriptionally activated precursor IgM^hi B-1a cells internalize surface BCR and differentiate into APCs.

**B-1a cells from CKO mice induce Tfh cells and GCs in wildtype recipients.** Unlike FOB and MZB cells, B-1a cells are poorly generated from the lymphoid progenitors in the BM[3,5,6]. To track the long-term function of CTLA-4-deficient B-1a cells in wild type (WT) recipients, we employ an IgH allotype chimera approach. This approach is based on the distinctive developmental properties of the B-1a and conventional B cells to generate B-cell chimeric animals[15,47,48]. Further, differing from the commonly used transfer systems, in which the irradiated adult animals are used as the recipients, we transfer B-1a cells into the non-irradiated IgH allotype-congenic newborn mice. This method allows the transferred B-1a cells to develop through

ontogeny in the normal host environment that most closely mimics the generation of B-1a cells during ontogeny.

To do that, we deplete the recipient B cells by treating 1-day-old CB.17 neonates (IgH[b]) with anti-IgM[b] monoclonal antibodies. On day 2, we reconstitute the treated CB.17 neonatal recipients with B-1a cells from adult allotype-congenic BALB/c mice (IgH[a]). The anti-IgM[b] antibody treatment continues for 6 weeks, after which the recipients are rested for 2–3 months during which they become B-cell chimera in which the host conventional B cells (IgH[b]) are generated from the endogenous BM, whereas the B-1a cells (IgH[a]) are reconstituted from the transferred, allotype-marked, self-replenishing B-1a (Supplementary Fig. 7A)[15,47,48]. With this approach, we transfer CTLA-4-deficient peritoneal B-1a cells from CKO mice (IgH[a]) into B-cell depleted congenic 2-day-old CB.17 neonates (IgH[b]). As controls, we transfer similarly treated CB.17 neonates with comparable numbers of CTLA-4-expressing peritoneal B-1a cells from CTLA-4 heterozygous mice (IgH[a]).

Examined after the recipients become chimeric adults, we found that both CD4 Tfh (PD1[hi] ICOS[+] GL7[+] CXCR5[+] BCL6[+]) and GC B cells (CD95[+] CD38[neg] GL7[+]) are induced in spleens of recipients that are reconstituted with CTLA-4-deficient peritoneal B-1a cells (Fig. 7a, b, Supplementary Fig. 7B). In contrast, neither Tfh cells nor GC B cells are induced in recipients reconstituted with CTLA-4-expressing peritoneal B-1a cells (Fig. 7a, b), even though like CTLA-4-deficient peritoneal B-1a cells, these cells also efficiently reconstitute B-1a populations in both spleen and PerC of the recipients (Supplementary Fig. 7C). In essence, induction of Tfh cells and GCs only proceeds when the recipients receive B-1a cells that do not express CTLA-4.

Further, CTLA-4-deficient B-1a cells not only induce Tfh cells in recipient spleens, TCR repertoires expressed by these Tfh cells are also similar to the repertoires expressed by Tfh cells in CKO mice. Thus, differing from non-Tfh T cells, which express a diverse TCRβ repertoire, the Tfh repertoires from both chimeric recipients and CKO mice are much less random and more repetitive (Fig. 7c, Supplementary Table 3), partly owing to both Tfh repertoires preferentially using TCR Vβ5, Vβ13, and Vβ31 genes (Supplementary Fig. 7D). In addition, for these Vβ genes, certain Vβ-Dβ-Jβ recombination sequences encoding particular CDR3 peptides are shared by Tfh repertoires from both CKO and chimeric mice (Supplementary Table 4). Our findings that the Tfh cells induced in the chimeric recipients express a similarly selected TCRβ repertoire as that expressed by Tfh cells in CKO mice strongly suggest that like Tfh cells induced in that chimeric mice reconstituted with CTLA-4-deficient B-1a cells, Tfh cells that spontaneously arise in CKO mice are also induced by CTLA-4-deficient B-1a cells in these animals.

Like peritoneal IgM[hi] B-1a cells, splenic IgM[hi] B-1a from CKO mice also induces Tfh cells and GC responses when transferred to and reconstituted in CB.17 recipients (Supplementary Fig. 7E). As demonstrated above, these CTLA-4-deficient IgM[hi] B-1a cells internalize surface IgM and their differentiated IgM[neg] B-1a decedents express not only the APC phenotype but also the gene programs of APC function. These differentiated IgM[neg] B-1a cells likely act as APCs that induce the Tfh cell differentiation. Supporting this idea, we find that transferring as few as 10[5] IgM[neg] B-1a cells from CKO mice efficiently induces Tfh cells and GC formation in the spleen of the recipients (Fig. 7b, Supplementary Fig. 7F).

Further, in spleens of recipients that are reconstituted with CTLA-4-deficient B-1a cells, ~40% of GC B cells express surface IgG1 (Fig. 7d, Supplementary Table 5). Using a monoclonal antibody that specifically detects IgG1 of IgHa-haplotype mice but does not react with IgHb haplotype[49], we further show that these IgG1-expressing GC B cells express donor a-haplotype

(IgG1a), indicating that they are derived from the transferred CTLA-4-deficient B-1a cells (IgH[a]), rather than the endogenous B cells from the CB.17 recipients (IgH[b]) (Fig. 7d). Therefore, CTLA-4-deficient B-1a cells not only induce GC responses but also directly give rise to GC B cells in the recipients.

**Reconstituted CTLA-4-deficient B-1a cells associate with FDCs.** B-1a cell development occurs during fetal and earlier neonatal life and largely ceases thereafter[13,50]. Although B-1a cells arise in spleen earlier in life, they comprise only a small percentage of splenic B cells in the adult spleen[51]. Their small number and lack of an exclusive marker make it difficult to identify their location in the normal spleen. Nevertheless, by using VH11 knock-in transgenic mice, Hardy and Hayakawa groups found that B-1a cells are localized centrally in the spleen follicles, where they expand in association with follicular dendritic cells (FDCs) during ontogeny[51].

Our findings here confirm and extend these findings. Thus, when transferred to and reconstituted in the non-irradiated recipients, CTLA-4-deficient IgMa[+] B-1a cells are located more centrally than the host IgD[hi] FOB cells in the follicles, where they are closely associated with the FDC network (Fig. 8, Supplementary Fig. 8A). As in the normal spleen, Hardy groups did not find the GC formation in spleens of VH11 knock-in transgenic mice. In contrast, we show that the GCs are formed in the spleen of recipients when are reconstituted with CTLA-4-deficient B-1a cells (Fig. 8). Furthermore, some of the donor IgMa[+] B-1a cells overlap with GC, where they interact with Tfh cells (Fig. 8, Supplementary Fig. 8B, C).

## Discussion

CD5[+] B-1a cells are considered as the evolutionarily primitive B cells. These B cells emerge during fetal development, express a repertoire enriched for autoreactive Ig and, play the essential and non-redundant roles in the immune system[1–3,12,17,18,52–54]. We report here that CTLA-4, a key T-cell regulator, is also expressed by B-1a cells and negatively regulates the function of these cells. This regulation, we further show, is indispensable for the maintenance of immune homeostasis. In essence, our findings unveil a new aspect of CTLA-4 immune function as a regulator of this unique B-cell subset and introduce it as an important immune-regulatory mechanism.

B-1a cells produce useful autoreactive natural antibodies[55,56]. This function, as we previously have shown, is enabled by their unique Ig repertoire that is selected by self-antigens during ontogeny[12]. The repertoire-defining mechanism, however, must prevent the BCR-mediated self-antigen presentation and high-affinity antibody-producing GC formation. Our studies identify CTLA-4 as an important player in this control process.

We show that B-1 cells selectively express CTLA-4 within the resting B-cell compartment. By specifically deleting CTLA-4 from B cells in CKO mice, we demonstrate that the CTLA-4-deficient B-1a cells become dysfunctional. That is, they upregulate epigenetic and transcriptional programs, show increased self-replenishment and express increased activation markers such as MHC II. Further, these activated cells internalize surface IgM and differentiate into APCs. Most importantly, when transferred to and reconstituted in IgH allotype-congenic recipients, these activated B-1a cells and their differentiated APC decedents are able to induce Tfh cells and GCs in recipient spleens.

Derived from the fetal development, B-1a cells appear earlier than FOB and MZB in the spleen where they undergo selection and self-replenishment after their development ceases[12]. By monitoring the splenic B-1a cells from the neonates to the adults, we show that B-1a *Ctla4* expression gradually increase over this

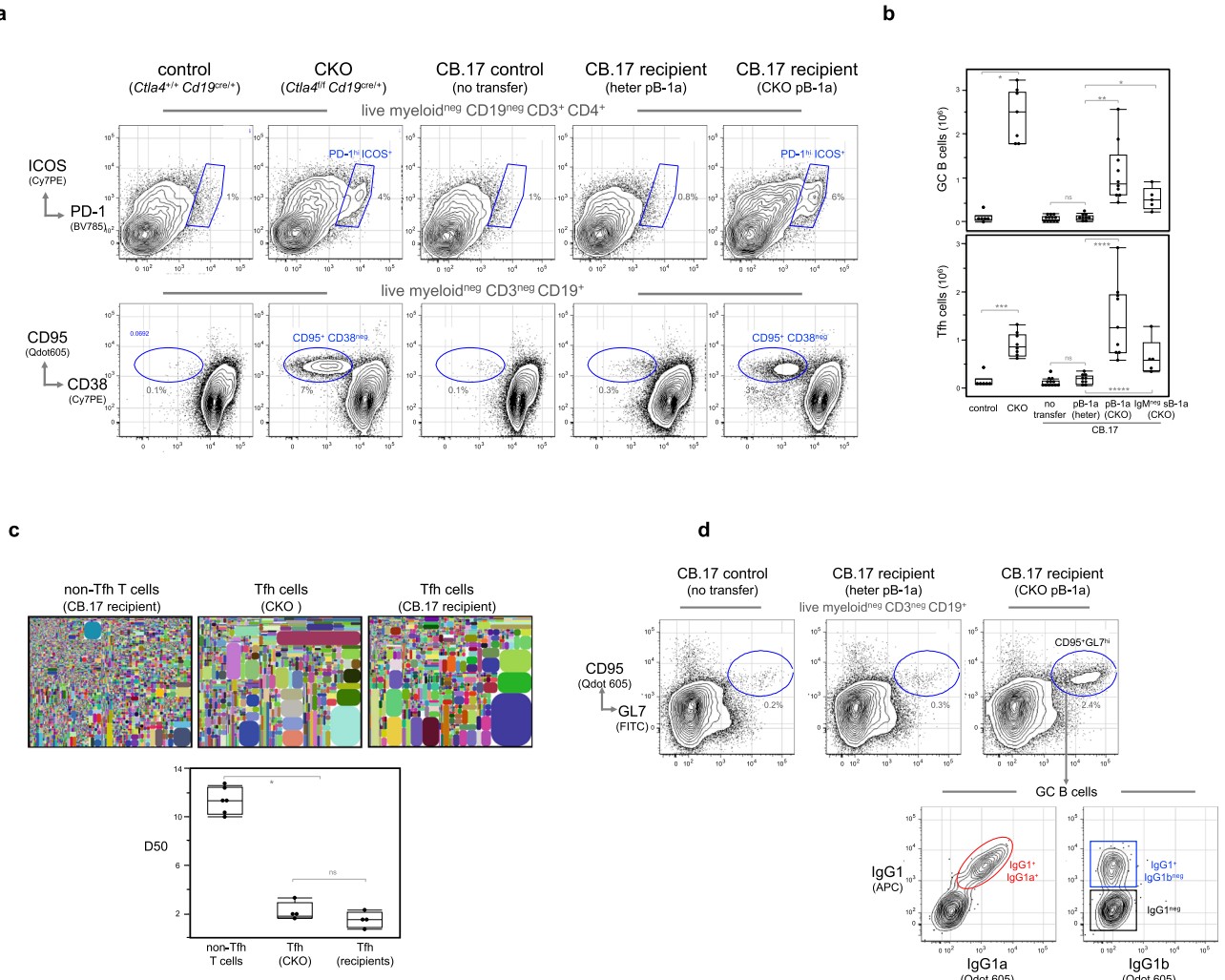

**Fig. 7 CTLA-4-deficient B-1a cells and their IgM^neg decedents induce Tfh cells and GCs in congenic recipients. a** Live CD4+ T cells and CD19+ B cells from spleens of indicated mice were gated to reveal Tfh (PD-1^hi ICOS+) and GC B (CD95^hi CD38^neg) cells, respectively. **b** Data summarizing the numbers of Tfh and GC B cells in spleens of indicated mouse groups is shown, $n = 7–10$ mice per group. Each dot represents data for an individual mouse, $*p <$ 0.003, $**p < 0.0002$, $***p < 0.002$, $****p < 0.0003$, $*****p < 0.004$, ns, not significant ($p < 0.2$), nonparametric Wilcoxon one-way test. About $5 \times 10^6$ of peritoneal B-1a (pB-1a) and $2 \times 10^5$ of IgM^neg splenic B-1a (sB-1a) cells were transferred. **c** TCRβ CDR3 tree-map plots (upper) illustrate the CDR3 nucleotide sequences expressed by non-Tfh T and Tfh cells from indicated mice. One representative plot from four or six independent T-cell samples is shown. Each rectangle in a given tree-map represents a unique CDR3 nucleotide sequence and the size of each rectangle denotes the relative frequency of an individual CDR3 sequence. The box plot summarizes the D50 metric analysis that quantifies TCRβ CDR3 nucleotide sequence diversity for each T-cell sample from indicated mouse group. Each dot represents data for a T-cell sample from an individual mouse, $n = 4–6$ T-cell samples per group, $*p < 0.006$, ns, not significant ($p < 0.2$), nonparametric Wilcoxon one-way test. Low D50 values are associated with less TCRβ CDR3 diversity. Sequence information for each T-cell sample is summarized in supplementary Table 3. **d** Live myeloid^neg CD3^neg CD19+ splenic B cells from indicated CB.17 mice were gated to reveal GC B cells (CD19+CD95+GL7+), which were further gated to show IgG1, IgG1b and IgG1a surface expression. Statistical summary for five independent experiments is shown in Supplementary Table 5. Box plots in **b**, **c**: box draws 75% (upper), 50% (center line) and 25% (down) quartile, the maxima and minima outliers are shown as top and bottom line, respectively.

peroid. This pattern is reminiscent of the gradual increase of recurring V(D)J sequences in the B-1a IgH repertoire during early ontogeny[12], raising questions as to whether the induction of CTLA-4 expression in B-1a is the result of B-1a cell selection during the normal development or tumor transformation. Consistent with this idea, we find that CTLA-4 expression is elevated in transformed B-1a cells in Eμ-TCL1 transgenic mice, in which B-1a cells develop into CLL-like tumors[30–32].

Naturally occurring Treg cells constitutively express high levels of CTLA-4, whereas other T cells induce CTLA-4 after they are activated. As a key coinhibitor, CTLA-4 shows extreme potency in negatively regulating T-cell function. Both fully CTLA-4-deficient ($Ctla4^{-/-}$) and CTLA-4 Treg knockout mice develop fatal

autoimmunity at earlier age, i.e., $Ctla4^{-/-}$ mice die by 3–4 weeks of age[26,27], whereas the CTLA-4 Treg knockout mice become inactive ~7 weeks[22]. Unlike T-cell CTLA-4-deficiency mice, CTLA-4 B-cell CKO mice live >1 year and do not develop the massive lymphoproliferation and multi-organ tissue destruction. Instead, they generate spontaneous GC responses that primarily occur in the spleen and elevate the autoantibody production as animals age. In later life (older than 7 months), some of these mice manifest the autoimmune pathology. The drastic differences for CTLA-4 deficiency in T versus B cells underscore the distinctive roles of these immune cells play in the immune system.

The GC responses that arise in the un-manipulated CTLA-4 B-cell CKO mice depend on CD4 T-cell and CD40 signaling, akin to

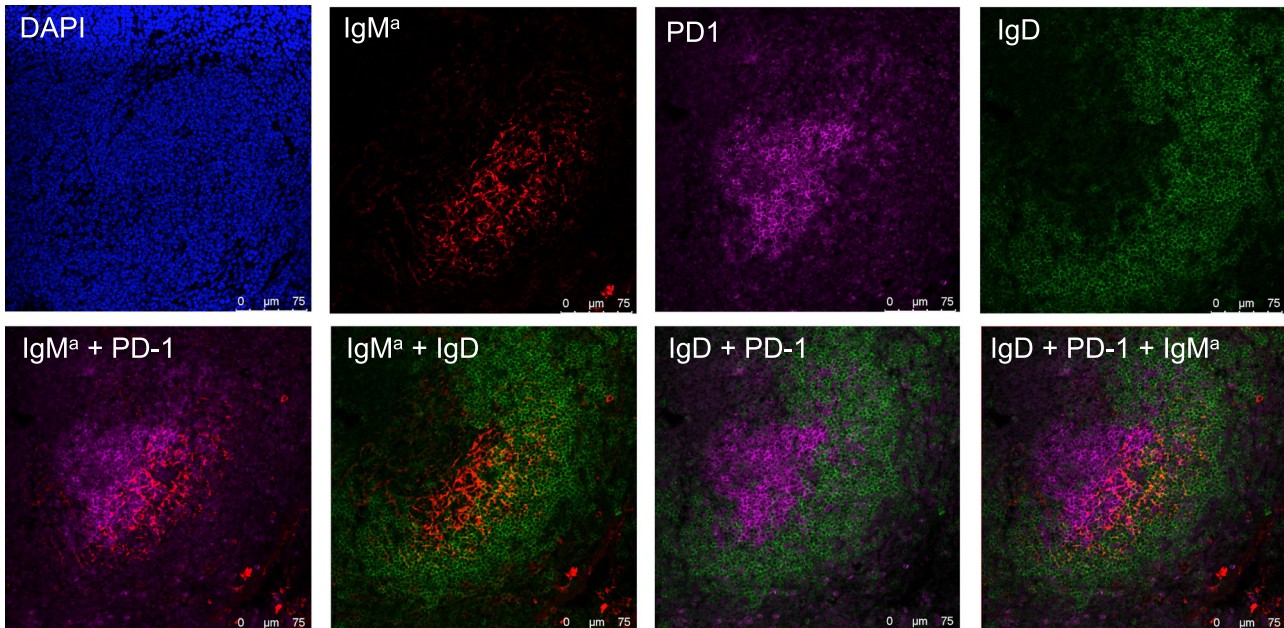

**Fig. 8 CTLA-4-deficient B-1a cells are localized in close association with FDCs in transfer recipient spleen.** Spleen sections of recipients that have been received peritoneal B-1a cells from CTLA-4 B-cell CKO mice were analyzed by immunofluorescence confocal microscopy. Representative immunofluorescence images of the section stained with indicated individual reagent are shown in the upper and images of two or three-color composites are shown in the bottom.

the GC responses induced by foreign antigens. Consistently, Tfh cells, which play crucial roles in the development of GC response[39,57], spontaneously arise in CKO mice. In foreign antigens-induced GC responses, APCs such as DCs usually capture and present antigens to naive CD4 T cells. This initial engagement instructs naive T cells to differentiate into Tfh cells[39]. No foreign antigens are introduced into CKO mice. Consistently, we find the splenic DC compartment in CKO mice remain inactive. Our studies show that unlike the foreign-antigen induced GC responses, Tfh cells in CKO mice are induced by the activated B-1a cells in these mice, and the induction is highly likely mediated by their differentiated APCs.

We show that B-1a cells in CKO mice upregulate epigenetic and transcriptional activation programs and the activated B-1a cells undergo further differentiation to APCs. They internalize surface IgM, and the resultant IgM$^{neg}$ B-1a cells start to express an APC phenotype (CD95$^+$ CD38$^{lo}$ GL7$^+$ PNA$^+$ CD150$^+$ csIg$^+$). Strikingly, this phenotype is also expressed by IgM$^{neg}$ IgD$^{neg}$ thymic B cells, which we show here express the B-1a phenotype and have been reported to function as the APCs that present self-antigens to developing T cells in the thymus[42]. The idea that these differentiated IgM$^{neg}$ B-1a cells are APCs is further supported by the finding that they upregulate MHC II and express comparable levels of H2-DM and H2-DO, the key players of MHC II processing and presentation, as light zone GC B cells, the well-known APCs in the GC. Moreover, not only expressing the APC phenotype, RNA-seq analysis showed that they also express the gene programs of APC function.

Notably, the CD95$^+$ CD38$^{lo}$ GL7$^+$ PNA$^+$ CD150$^+$ csIg$^+$ IgM$^{neg}$ B-1a cells are the APCs prior to GC formation. Although they share some of the GC B-cell phenotype (CD95$^+$ CD38$^{neg}$ GL7$^{hi}$ PNA$^{hi}$ CD150$^+$ csIg$^+$), they have not yet fully differentiated to GC B cells, which further down regulate CD38 and upregulate GL7 and PNA. We call these cells pre-GC APCs to distinguish them from the APCs that present antigens to T cells in the GC. In essence, we identify an APC population, which is derived from the CTLA-4-deficient IgM$^{hi}$ B-1a cells that have

internalized surface IgM but have not yet turned off surface CD5 expression. These APCs share the same phenotype as thymic IgM$^{neg}$ IgD$^{neg}$ B-1a cell, the APCs in the thymus; but have not yet differentiated to APCs within the fully developed GCs.

B cells become potent APCs after the BCR-mediated internalization of the antigen. Although their ability to present antigen to CD4 T cells is less efficient than DCs or macrophages when they take up the antigens non-specifically, their antigen-presenting efficiency increases up to 10$^4$ folds when antigens are recognized through BCR[58]. The intracellular IgM that we detected in IgM$^{neg}$ B-1a cells of CKO mice are not the secreted IgM from the IgM-secreting cells and hence most likely represent the internalized IgM (BCR) from the surface. Accordingly, RNA-seq shows that IgM$^{neg}$ B-1a cells do not express a PC signature; instead, they express the gene programs that are associated with APC function.

Most importantly, in the functional assay, we show that when transferred to and reconstituted in normal congenic recipients, both CTLA-4-deficient IgM$^{hi}$ B-1a and their differentiated IgM$^{neg}$ decedents induce Tfh cells and GC responses in recipient spleen. Further, these cells not only induce GC formation but also directly develop into the class-switched GC B cells. As B-1a repertoire is enriched for the self-reactive Ig, it is conceivable that the internalized BCR in IgM$^{neg}$ B-1a cells may facilitate trafficking, processing, and presenting their binding self-antigens to naive CD4 T cells, thereby induce them to differentiate to Tfh cells.

Further, we show that Tfh cells that are induced in the recipients express a similarly selected TCRβ repertoire as the Tfh cells that spontaneously arise in CKO mice. Both repertoires are much less random and more repetitive than non-Tfh T-cell repertoire, and are shared with certain Vβ-Dβ-Jβ recombination sequences that encode particular CDR3 peptides. Our finding that Tfh cells from different mice under different experimental settings (CKO and chimeric mice) express the similarly selected TCRβ repertoires not only underscores a common mechanism that these Tfh cells are induced by CTLA-4-deficient B-1a cells, it also

suggests that these Tfh cells are induced by some common self-antigens.

These Tfh cells are derived from naive CD4 T cells, raising the questions as for how are these self-reactive CD4 T cells generated and released into the periphery? It is well held that the self-reactive T cells in the thymus are deleted by negative selection, referred as central tolerance. Intriguingly, emerging studies have shown that thymic B cells also participate in the central tolerance induction[43–45]. IgM[neg] IgD[neg] thymic B cells, in particular, have been recently shown to function as APCs presenting self-antigen to thymocytes[42]. We further show that most IgM[neg] IgD[neg] thymic B cells express CD5[+]CD43[+] B-1a phenotype. Whether thymic B-1a cells play roles in the selection of T-cell repertoire remain as open questions.

In both CKO and chimeric mice reconstituted with CTLA-4-deficient B-1a cells, spleen is the principle lymphoid organ in which Tfh cells and GC responses occur. Nevertheless, we show that loss of CTLA-4 affects both splenic and peritoneal B-1a cells in CKO mice. Both populations show an increased percentage of BrdU-incorporated cells (BrdU[+]), indicating an increased self-replenishment compare to their control counterparts. As a result, the percentage and the numbers of both splenic and peritoneal B-1a cells in CKO mice are elevated. Most importantly, when transferred to and reconstituted in normal congenic recipients, splenic and peritoneal IgM[hi] B-1a cells from CKO mice show similar functional consequence in recipients. That is, each population not only reconstitutes B-1a cells in both spleen and PerC of the recipients, it also induces Tfh cells and GC responses in spleens of these mice.

Thus, although splenic B-1a express lower levels of CTLA-4 than peritoneal B-1a cells, both populations in CKO mice are dysfunctional in that they are able to induce Tfh cells and GC response when reconstituted in normal congenic recipients. Intriguingly, the CTLA-4 expression pattern also occurs in splenic and peritoneal B-1a cells of other mice including $Tcrb^{-/-}$ $Tcrd^{-/-}$ and Eμ-TCL1 Tg mice. Whether the difference in CTLA-4 expression between splenic B-1a cells versus peritoneal B-1a cells reflects their distinct activation status remains unclear. Consistent with the notion that B-1a cells undergo self-replenishment in the spleen[12,35,36], our in vivo BrdU incorporation studies show that in both CKO and control mice, splenic B-1a cells always have much higher percentage of BrdU[+] cells than peritoneal B-1a cells.

As both self-antigen and BCR signaling are indispensable for the B-1a generation[1,2], it is conceivable that B-1a self-replenishment is fueled by constant BCR engagement with self-antigen in the spleen. Intriguingly, Hardy and Hayakawa et al.[51] reported that PtC-specific B-1a cells are located centrally in the spleen follicles, where they expand in association with FDCs. Consistent with this key finding, we also show that when reconstituted in the recipient spleens, CTLA-4-deficient B-1a cells locate more centrally than FOB cells in the follicles, where they are closely associated with FDCs.

Acting as the antigen depot, FDCs present antigens both to naive B cells as they survey primary follicles and to GC B cells when they compete for antigen within GCs[59]. We postulate that in normal resting state, B-1a cells are constitutively stimulated with self-antigens loaded by FDCs. Such interaction is sufficient to stimulate B-1a cells to undergo self-replenishment; however, it should be under tight control to avoid over-activating these cells. CTLA-4 expression by B-1a cells may function to modulate their BCR interaction with self-antigens loaded by FDCs. In fact, we find that when B-1a cells lose CTLA-4, they undergo epigenetic and transcriptional reprogramming and upregulate genes of transcription activator, cell cycle and the PI3K/Jak-Stat BCR signaling pathway. As a result, they show increased self-replenishment and express activation markers.

Studies with T cells have shown that CTLA-4 engagement with its ligands delivers a negative signaling[25]. Whether CTLA-4 regulates B-1a function through this cell-intrinsic mechanism remains unsolved in this study. We find both splenic and peritoneal B-1a cells express CD80 and CD86. In particular, levels of CD80, the strongest CTLA-4 ligand, are surprisingly high on B-1a cells. Co-expression of CTLA-4 and its ligands, particular CD80, by B-1a cells suggests that CTLA-4-B7 interaction can occur in multiple ways for B-1a cells, i.e., it can occur in a single B-1a cell or may require a two-cell interaction between CTLA-4-expressing B-1a and another B7-expressing B-1a cells. Therefore, although B-1a cells express low levels of CTLA-4, their CTLA-4 has close access to its ligands, which may be uniquely involved in CTLA-4 regulation of B-1a cell activation status, e.g., by fine-tuning the BCR signaling.

Taken together, studies presented here have elucidated a new aspect of the function of CTLA-4 in the maintenance of immune homeostasis. This function, as we demonstrated, is mediated by regulating B-1a cells, a B-cell population that arises during fetal development and expresses an Ig repertoire enriched for auto-reactivity. Further, our findings offer potential insights into the adverse autoimmunity associated with "CTLA-4 blockade" cancer immunotherapy[60,61]. Anti-CTLA-4 antibody treatment to block CTLA-4 signaling in T cells shows promising effects in advanced cancer patients[60]. However, clinical studies may also find serious autoimmune pathologies associated with this therapy. The underlying mechanisms have yet to be fully understood. However, since CTLA-4 is likely expressed by human B-1a cells, our studies suggest that the dysfunction of human B-1a if/when they loss CTLA-4 regulation may contribute to the autoimmunity observed in patients receiving CTLA-4 blockade immunotherapy.

## Methods

**Mice.** BALB/c (IgH[a]), CB.17 (IgH[b]), C57BL/6J, $Tcrb^{-/-}$ $Tcrd^{-/-}$ (C57BL/6J) and homozygous CD19cre mice ($Cd19^{tm1(cre)cgn}$) are purchased from Jackson laboratory. Eμ-TCL1 (C57BL/6J) transgenic mice are generously provided by Dr. Thomas Kipps at the UC San Diego Moores Cancer Center. BALB/c mice expressing a floxed CTLA-4 gene ($Ctla4^{fl/fl}$) were generously provided by Dr. Shimon Sakaguchi (Osaka University). Control mice ($Ctla4^{+/+}$ $Cd19^{cre/+}$) were obtained by mating CD19cre mice with BALB/c mice. To obtain the CTLA-4 B-cell CKO mice ($Ctla4^{fl/fl}$ $Cd19^{cre/+}$), $Ctla4^{fl/fl}$ mice were bred with $Ctla4^{fl/+}$ $Cd19^{cre/+}$ F1 mice and their progenies are genotyped by PCR using primers that detect the mutant or WT allele of $Ctla4$ and $Cd19$ gene. Primer information is shown in Supplementary Table 6. To obtain the CTLA-4 B-cell CKO and heterozygous ($Ctla4^{fl/+}$ $Cd19^{cre/+}$) mice that express IgH[a] allotype, mice blood was collected and analyzed by FACS using anti-IgM[b]-FITC (clone AF6-78, BD,cat#553520), anti-IgM[a]-PE(clone DS-1, BD, cat#553517). Mice were maintained on a 12h-light/dark cycle (light on at 7 am and off at 7 pm) and at the temperature of 68-79°F with 30–70% humidity in specific pathogen-free facilities. The animal care was conducted in accordance with the guidelines of Stanford Veterinary Service Center (VSC). All animal experiments were performed following the protocols approved by VSC.

**In vivo BrdU incorporation assay.** Two or 3-month-old CTLA-4 B-cell CKO mice ($Ctla4^{fl/fl}$ $Cd19^{cre/+}$) or control mice ($Ctla4^{+/+}$ $Cd19^{cre/+}$) were fed with water containing 0.8 mg/ml BrdU and 1% sucrose for 6 days. Mice were then killed, spleen, and PerC cells were analyzed by FACS.

**In vivo antibody treatment.** To deplete CD4[+] T cells or block CD40 signaling in vivo, CTLA-4 B-cell CKO mice were i.p. injected with 100 μg of anti-mouse CD4 antibody (clone GK1.4, Biolegend) or anti-mouse CD40L (clone MR1, Biolegend), respectively, twice per week for 5 weeks. As control groups, age and gender-matched CTLA-4 B-cell CKO mice were injected with 100 μg of IgG isotype control antibody (clone SHG-1, Biolegend).

**IgH allotype-congenic chimeric mice.** One-day-old newborn CB.17 (IgH[b]) mice were treated with 100 μg of anti-IgM[b] monoclonal antibodies (clone AF6-78, Biolegend). The next day, anti-IgM[b]-treated mice were i.p. transferred with peritoneal or splenic B-1a cells from CTLA-4 B-cell CKO ($Ctla4^{fl/fl}$ $Cd19^{cre/+}$) or heterozygous mice ($Ctla4^{fl/+}$ $Cd19^{cre/+}$) that express IgH[a] allotype. Anti-IgM[b] antibody treatment continues for 6 weeks with twice injections (100 μg/time) per week. After the last antibody injection, recipients were rested for 2–3 months to allow the B-cell reconstitution and become B-cell chimeric animals.

**qRT-PCR**. qRT-PCR was performed using CellsDirect One-Step qRT-PCR kit (ThermoFisher Scientific) following their instruction. In brief, phenotypically defined B-cell subsets were directly sorted into 96 wells containing lysis buffer and Superase-In (Ambion) with $10^3$ cells per well. The lysed samples were analyzed using TaqMan Gene Expression Assays with probes that detect the targeted genes (Applied Biosystems). *Ctla4*, Mm00486849_m1; *Plor2a*, Mm00839502_m1; *Bcl6*, Mm00477633_m1; *Aicda*, Mm00507774_m1. Real-time PCR is performed with 7900HT system (Stratagene MX3005).

**Mouse serum Ig analysis**. Levels of total IgM, IgG, and IgE in sera were measured using Mouse IgM, IgG total, and IgE ELISA Ready-SET-Go kits (eBioscience). To measure the levels of IgG anti-dsDNA antibodies, sera were analyzed with an ELISA kit (LBIS, Japan) following the manufacturer's protocol. To measure the IgE anti-dsDNA antibodies, we used the same kit except that an HRP-conjugated anti-mouse IgE (SouthernBiotech) was used. Levels of IgG anti-ANA antibodies were measured with a HEp-2 kit (Orgentec, Germany). The HEp-2 slides were incubated with mouse sera and stained with 1 μg/ml of Alexa488-goat anti-mouse IgG(H+L) (Life technologies). To measure anti-RF antibody levels, sera were analyzed with ELISA kit (LBIS, Japan) following the manufacture's instruction. To measure the anti-glycan autoantibody level, carbohydrate antigens were dissolved in phosphate-buffered saline (PBS) and spotted onto SuperEpoxy 2 Protein slides (ArrayIt Corporation, Sunnyvale, CA) in triplicate for each antigen preparation using microarray printer PixSys 5500C (Cartesian Technologies, Irvine, CA). The printed microarray slides were then washed with PBS and blocked with 1% bovine serum albumin–PBS at RT for 30 min. They were then incubated with serum antibodies at 1:100 dilutions at RT for 1 hour followed by washing and then incubated with Cy5-tagged goat anti-mouse IgG (2.0 μg/ml) and Alexa594-tagged goat anti-mouse IgM (2.0 μg/ml) secondary antibodies at RT for 30 min. The stained slides were rinsed five times and spin-dried at room temperature before scanning for fluorescent signals. The ScanArray5000A Microarray Scanner (PerkinElmer Life Science) was used to scan the stained microarrays and the mean fluorescent intensities of each microspot were calculated using ScanArray Express software (version 4.0, PerkinElmer Life Science). Anti-glycan autoantibody profiles of each sample were graphically presented as overlay plots using JMP genomics statistical software (JMP® Pro 14.1.0, SAS Institute) with the mean values of triplicate microspots of each antigen preparation in the y axis and corresponding autoantigen preparations in the x axis. Information of the glycan antigens in carbohydrate microarray is shown in Supplementary Table 7.

**FACS analysis and sorting**. To obtain the thymocytes, the connective tissues attached to thymic lobes were carefully peeled off to remove the parathymic LNs. Single-cell suspension from the spleen, LN, peritoneal cavity and thymus was prepared. About $1.25 \times 10^6$ cells were incubated with LIVE/DEAD Aqua (Invitrogen), washed, and incubated with unconjugated anti-CD16/CD32 (FcγRII/III) mAb (2.0 μg/ml) to block Fc-receptors. Cells were then stained on ice for 20 min with a "cocktail" of fluorochrome-conjugated antibodies. After washing, cells were stained with Streptavidin-Qdot 605 (Invitrogen). For intracellular staining, after the surface staining, cells were fixed and permeablized with Foxp3/transcription factor staining buffer set (cat#00-5523-00, ThermoFisher, Inc.) and then were stained with fluorochrome-conjugated antibodies. Each fluorochrome-conjugated antibody is used at 1:100 dilution in 200 μl staining volume. BrdU staining was performed using BrdU flow kit (BD Bioscience). Before staining thymic B cells, single-cell suspension cells from thymus were first enriched with B cells by negative selection using mouse pan-B-cell isolation kit (StemCell Technology). For FACS sorting of splenic B-1a cells, total splenic cells were enriched for B cells by negative selection using mouse pan-B-cell isolation kit (StemCell Technology). For FACS soring of PCs, splenic cells were first enriched for CD138$^+$ B cells by positive selection using mouse CD138-positive selection kit (StemCell Technology). After the enrichment, cells were then stained with surface makers. Cells were analyzed or sorted on FACSAria IIu (BD Biosciences) at Stanford Shared FACS Facility. FACS data were required using FACS Diva Software (version 8.0, BD Bioscience) and analyzed using FlowJo (version 9.7, Treestar).

**Antibodies**. Fluorochrome-conjugated antibodies include: anti-CD21-FITC (clone 7E9, Biolegend, cat#123408), anti-IAd-FITC(clone AMS-32.1, BD, cat#553547), anti-GL7-FITC (clone GL7, BD, cat553666), anti-IgMb-FITC (clone AF6-78, BD, cat#553520), anti-IgMa-BV785(clone DS-1, Biolegend, cat#743890), anti-CD43-PE (clone S11, Biolegend, cat#143206), anti-CD138-PE (clone 281-2, Biolegend, cat#142504), anti-CD267-APC (clone eBioBF10-3, eBioscience, cat#17-5942-82), anti-CD38-Alexa488 (clone 90, Biolegend, cat#102714), anti- H2-M-Alexa488 (clone 2E5A, BD, cat# 552405), anti-CD5-PE-Cy5 (clone 53-7.3, Biolegend, cat#100610), anti-CD19-PE-Cy5.5 (clone ID3, Invitrogen, cat#35-0193-82), anti-IgG1-PE-Cy7(clone RMG1-1, Biolegend, cat#406614), anti-ICOS-PE-Cy7(clone C398.4A, Biolegend, cat#313520), anti-CD150-PE-Cy7(clone TC150-12F12.2, Biolegend, cat#115913), anti-IgM-PE-Cy7(clone R6-60.2, BD,cat#552867), anti-B220-Alexa700(clone RA3-6B2, Biolegend, cat#103232), anti-CD3-Alexa700(clone 145-2C11, Biolegend, cat#100216), anti-IgM-APC (clone RMM1, Biolegend, cat#406509), anti-Bcl6-APC (clone 7D1, Biolegend, cat#648305), anti-Ki67-Alexa647 (clone 11F6, Biolegend, cat#151206), anti-CTLA-4-PE(clone UC10-4F10-

11, BD, cat#553720), PE Hamster IgG1 isotype control (clone A19.3, BD, cat#553972), anti-CTLA-4-APC (clone UC10-4F10-11, BD, cat#564331) anti-Foxp3-APC (clone FJK-16S, eBioscience, cat#17-5773-80), anti-Foxp3-PE (clone FJK-16S, eBioscience, cat#12-5773-82), anti-IgD-APC-Cy7(clone 11-26c.2a, Biolegend, cat#405716), anti-CD4-APC-Cy7(clone GK1.5, BD,cat#552051), anti-CD23-biotin (clone, B3B4, Biolegend, cat#101604), anti-CD80-biotin (clone 16-10A, BD, cat#553767), anti-CD86-biotin (clone GL-1, Biolegend, cat#105004), anti-CD95-Qdot605 (clone SA367H8, Biolegend, cat#152612), anti-IgG1a-biotin (clone 10.9, BD, cat#553500), anti-IgG1b-biotin (clone B68.2, BD, cat#553533), anti-CXCR5-Brilliant BV605 (clone L138D7, Biolegend, cat#145513), anti-CD11b-PB (clone M1/70, Biolegend, cat#101224), anti-Gr-1-PB(clone RB6-8C5, Biolegend, ca#108430), anti-TCRαβ-PB(clone H57, Invitrogen,cat#HM3628), anti-CD11c-PB (clone N418, Biolegend, cat#117322), anti-CD3ε-PB (clone 145-2C11, Biolegend, cat#100334), anti-F4/80-PB (clone BM8, Biolegend, cat#123124), anti-CD279 (PD-1)-Brilliant violet 785 (clone 29F.1A12, Biolegend, cat#135225), anti-CD25-PE (clone 7D4, BD, cat#558642). LEAF purified anti-CD154 (clone MR1, Biolegend, cat#106508), anti-CD4 (clone GK1.5, Biolegend, cat#100442), IgG isotype control (clone SHG-1, Biolegend, cat#70647), anti-IgM$^b$ (clone AF6-78, customer produced by Biolegend).

**Mouse IgH and TCRβ repertoire analysis**. About $1–2 \times 10^4$ of phenotypically defined B cells and T cells were sorted and RNA was extracted using RNeasy plus Micro Kits (Qiagen) following the manufacturer's instructions. RNA sample was used for amplicon-rescued multiplex PCR using primers provided by iRepertoire following the procedures as described previously[12]. In brief, cDNA was reverse transcribed from total RNA sample using mixture of forward $V_H$ and reverse $C_H$ primers and reagents from the OneStep RT-PCR kit (Qiagen, Valencia, CA). The RT-PCR1 condition for BCR is: 50 °C, 60 min; 95 °C, 15 min; 94 °C, 30 s, 63 °C, 5 min, 72 °C, 45 s, for 10 cycles; 94 °C, 30 s, 72 °C, 3 min, for 10 cycles; 72 °C, 15 min. The RT-PCR1 condition for TCR is: 50 °C, 60 min; 95 °C, 15 min; 94 °C, 30 s, 60 °C, 5 min, 72 °C, 45 s, for 10 cycles; 94 °C, 30 s, 72 °C, 3 min, for 10 cycles; 72 °C, 15 min. After the first-round of RT-PCR, primers were removed by Exonuclease I digestion at 37 °C for 30 min (New England Biolabs, lpswich, MA). Then 2 μl of the first-round RT-PCR products were used as templates for the second round of amplification using communal primers and reagents from the Multiplex PCR kit (Qiagen). The second round PCR was performed as: 95 °C, 3 min; 94 °C, 30 s, 72 °C, 90 s, for 30 cycles; 72 °C, 15 min. About 400-bp long PCR products were run on 2% agarose gels and purified using a gel extraction kit (Qiagen). The IgH libraries were pooled and sequenced by Illumina paired-end sequencing (Illumina MiSeq platform). The sequence information for all primers used for the library preparation can be found in US Patent Office (US9012148).

Sequence reads were de-multiplexed according to barcode sequences at the 5'-end of reads from the IgH constant region. Reads were then trimmed according to their base qualities with a 2-base sliding window, if either quality value in this window is lower than 20, this sequence stretches from the window to 3'-end were trimmed out from the original read. Trimmed pair-end reads were joined together through overlapping alignment with a modified Needleman-Wunsch algorithm. If paired forward and reverse reads in the overlapping region were not perfectly matched, both forward and reverse reads were thrown out without further consideration. The merged reads were mapped using a Smith-Waterman algorithm to germline V, D, J, and C reference sequences downloaded from the IMGT website (Lefranc, 2003). To define the CDR3 region, the position of CDR3 boundaries of reference sequences from the IMGT database was migrated onto reads through mapping results and the resulting CDR3 regions were extracted and translated into amino acids.

The forward $V_H$ primers used to amplify expressed IgH genes are located at the IgH framework region 2. To avoid primers interfering with the mutation analysis, the variable region stretching from the beginning of the CDR2 to the beginning to the CDR3 was examined for mismatches between the sequence read and the best-aligned germline reference sequence. To eliminate the impact of sequencing error on this calculation, only sequence reads with more than four copies were included in the mutation calculation.

D50 is a measurement of the diversity of an immune repertoire of J individuals (total number of CDR3s, Eq. (1)) composed of S distinct CDR3s in a ranked dominance configuration, where $r_i$ is the abundance of the most abundant CDR3: $r_1$ is the abundance of the most abundant CDR3, $r_2$ is the abundance of the second most abundant CDR3, and so on. C is the minimum number of distinct CDR3s amounting to ≥50% total sequencing reads. D50 is given by Eq. (2)

$$\text{Assume that } \underbrace{r_1 \geq r_2 \ldots \geq r_{i+1} \ldots r_s}_{s}, \sum_{i=1}^{S} r_i = J \tag{1}$$

$$if \sum_{i=1}^{C} r_i \geq J/2 \text{ and } \sum_{i=1}^{C-1} r_i < J/2, \ D50 = \frac{C}{S} \times 100 \tag{2}$$

To draw the TCRβ CDR3 tree-map for each sequencing sample, the entire rectangle is first divided into a set of rectangles with each rectangle corresponding to a distinct Vβ gene segment. The size of the individual rectangle is proportional to the relative frequency of sequence expressing a given Vβ gene. The rectangles are ordered based on area from large at the bottom right to small at the top left. Each Vβ rectangle is then further divided into a set of Vβ-Jβ rectangles with each

rectangle corresponding to a distinct Vβ-Jβ combination. The size of the Vβ-Jβ rectangle is proportional to the relative frequency for a given Vβ-Jβ combination sequence. Vβ-Jβ rectangles are ordered based on area from large at the bottom right to small at the top left. Similarly, each Vβ-Jβ rectangles is further divided into a set of Vβ-Jβ-CDR3 rectangles with each rectangle representing a distinct Vβ-Jβ-CDR3 combination. The size of the particular Vβ-Jβ-CDR3 rectangle is proportional to the relative frequency for each Vβ-Jβ-CDR3 combination sequence. Therefore, each rectangle drawn in the map represents an individual CDR3 nucleotide sequence. In order to distinguish neighboring rectangles, corners of each rectangle are rounded and each rectangle are colored randomly.

**RNA-seq analysis**. RNA was isolated from $1-3 \times 10^5$ sorted cells using RNeasy plus Micro Kits (Qiagen), according to the manufacturer's instructions. RNA concentration and integrity number were quantified by RNA quality control analysis conducted by Stanford Protein and Nucleic Acid Facility. About 30 ng of RNA per sample was used for preparing sequence library. In brief, ribosome RNA was removed from the RNA material using Ribo-zero magnetic gold kit (illumine). After the ribo-minus RNA was fragmented, cDNA was synthesized using super-script double-strand cDNA synthesis kit (Invitrogen). The cDNA samples were then end repaired, ligated with illumine adaptor using NEB next ultra II end repair/dA-tailing and ligation module (New England Biolabs), and amplified with illu-mine sequencing primer. The amplified cDNA libraries were sequenced by Illumina Hiseq 4000 at the Genome Sequencing Service Center by Stanford Center for Genomics and Personalized Medicine Sequencing.

We aligned RNA-seq reads to the mouse reference genome (mm9) using TopHat (version 2.0.13) with the transcript annotation supplied. Both the mouse reference genome and transcript annotation were downloaded from illumina iGenomes. We then assigned the mapped reads to gene using Python package HTseq (Python 2.7, HTseq version 0.6.0), with the default union-counting mode. To perform differential gene expression analysis, we applied DESeq2 R package (R 3.3.1, DEseq 2 version 1.12.4), with an adjusted $P$ value of 0.05 as the cutoff. We performed gene set annotation enrichment analysis using The Database for Annotation, Visualization and Integrated Discovery (DAVID) functional annotation tool web server. Terms with a false discovery rate (FDR) of 10% were considered significant. We then adapted GOPlot R package (version 1.0.2) to visualize the gene set annotation enrichment analysis results.

**Immunofluorescence confocal microscopy**. Spleen was fixed in 4% paraformaldehyde for 2 h at 4 °C, then incubated in 10% and 20% sucrose sequentially at 4 °C for 30 mins. After incubation in 30% sucrose overnight, spleen was embedded in OCT compound (Lab-Tek Products, Naperville, IL). The frozen spleen was cut in 7 μm section. The cryo-sections were washed with PBS, blocked with normal rat serum and Fc block antibody (2.4G2 clone) for 1 h, and stained with fluorochrome-labeled antibodies. After washing with PBS, sections were stained with DAPI and mount using Fluoromount-G (Southern Biotech). Images were required with Leica SP5 confocal (Wetzlar, Germany) and analyzed by LAS AF 2.7.9723.3 software of Leica Microsystem.

**Statistics and reproducibility**. Statistical analysis was performed suing JMP genomics statistical software (JMP® Pro 14.1.0) (SAS Institute). Statistical comparisons between groups were performed using nonparametric Wilcoxon one-way test. The representative data shown in this study represents the result from at least four independent experiments with similar results.

**Reporting summary**. Further information on research design is available in the Nature Research Reporting Summary linked to this article.

## Data availability
IgH and TCRβ sequence data sets generated in this study have been deposited at NCBI Sequence Read Archive (SRA) under the accession code RPJNA678122. The RNA-seq data set has been deposited at Gene Expression Omnibus (GEO) under the accession code GSE161409. All other data are available within the article and Supplementary information files, or from the corresponding authors on reasonable request. Source data are provided with this paper.

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

## Acknowledgements

We thank David Parks, Wayne Moore, Ometa Herman, Lisa Nichols, and other staffs in Stanford Shared FACS facility (NIH S10 Shared Instrument Grant) for help with FACS analysis and sorting; Kitty Lee and the Cell Sciences Imaging Facility for Immuno-fluorescence confocal microscopy analysis (S10RR02557401 from the National Center for Research Resources). We thank Dr. Thomas J. Kipps at the UC San Diego Moores Cancer Center for providing Eμ-TCL1 transgenic mice and Dr. Elizabeth D. Mellins at Stanford University for giving anti-mouse H2-DO antibody. Tissue anatomic pathology analysis was performed by Animal Histology Services in the Department of Comparative Medicine at Stanford. RNA-seq sequencing was performed in Genome Sequen-cing Service Center by Stanford Center for Genomics and the Personalized Medi-cine Sequencing Center for RNA-seq analysis (NIH S10OD020141). The repertoire studies were supported by HudsonAlpha Institute for Biotechnology and iRepertoire (J.H.). Autoantibody analysis was supported by NSFC 91942302 and National Key R&D Project 2019 YFE0100600 (J.-Y.W.). The carbohydrate microarray work was supported by NIH grants 1R21AI124068 (D.W.), 1R21DA046144 (D.W.), and CDMRP grant PR170128 (D.W.). We acknowledge the Kabat Collection of Carbohydrate Antigens at SRI International for a panel of carbohydrate antigens that were applied in this study. We thank Dr. Aaron B. Kantor and Dr. Paula Kavathas (Yale University) for critical reading of the manuscript. Work was supported by NIH R01 AI128839-01.

## Author contributions

Y.Y. conceived of and designed the project; Y.Y., G.Q., and J.V.Y. performed the experiments; X.L. and Z.G. performed bioinformatics RNA-sequencing and provided data; Z.M. and Y.C. prepared the cDNA libraries for RNA-seq analysis; C.W., Q.Y., and M.B-S. performed the IgH and TCRβ repertoire analysis and provided data; J.H. supervised IgH and TCRβ repertoire study; R.H., Q.M., and J-Y.W. conducted the assays testing the anti-dsDNA, ANA, and RF antibodies in mice sera and provided data; J.B.W. and S.S. provided *Ctla4*-floxed mice and expertize in analyzing Treg and Tfh cells; C.T. and L-X.W. provided glycan antigens; D.W. conducted IgM and IgG anti-glycan pro-filing assay and provided data; J.G.V. conducted immune pathology analysis and pro-vided data; M.P.S. supervised the RNA-seq analysis; Y.Y. and L.A.H. wrote the manuscript.

## Competing interests

C.W. and J.H. are co-founders of iRepertoire. M.P.S. is cofounder and a member of the scientific advisory board of Personalis, Qbio, January, SensOmics, Protos, Mirvie, Ora-lome. M.P.S. is on the scientific advisory board of Danaher, Genapsys, and Jupiter. The other authors declare no competing interests.
