## [Peer Review File · Nature Communications]

REVIEWER COMMENTS

Reviewer #1 (Remarks to the Author):

The authors have responded to previous critique by providing a convincing and comprehensive analysis of mice lacking CTLA-4 specifically in B cells (or B-1a cells). The manuscript is clear, the data quality high and the findings should be of interest to a wide audience, given the importance of CTLA4 as a clinical target. While the mechanisms of CTLA4-mediated autoantibody production and spontaneous germinal center formation remain to be elucidated, the manuscript provides a comprehensive first description of CTLA4 as a regulator of B cell immunity. I have no further suggestions for improvement.

Reviewer #3 (Remarks to the Author):

As I stated in my previous report, given the exceptional groundbreaking nature of the authors' observations this manuscript will likely be very controversial and attract a very high level of scrutiny. I am still in disbelief that a very minor level of CTLA-4 on a small proportion of B-1a cells can have such dramatic functional impact *in vivo*. Although the authors made some efforts to suggest a putative underlying mechanism, it remains unconvincing how such a mechanism may operate. However, as I mentioned before, the functional data provided are convincing and conclusive, making the functional impact of CTLA-4 loss B-1a cells hard to dispute.

My only request, at this stage, is linked to the need to be completely open with the data. The authors' have been unable or unwilling to provide FACS data to document CTLA-4 expression on all B1-a populations used in the different experiments. I feel this should not be a difficult request.

1. A dot plot (not an histogram) should be included in the main figures to show the level of CTLA-4 expression on the cell populations displayed in Figure 1c. The authors provided the gating strategy but not the requested CTLA-4 dotplot. The dotplot is important to assess what are the characteristics of the cells that are marginally expressing more CTLA-4.

2. As the authors use experimentally sB-1a cells from CTLA4^{+/+} (control), CTLA4^{fl/+} (heter), and CTLA4^{fl/fl} (mutant), FACS dotplots (not histograms) showing the level of CTLA-4 on those three cell populations should be added to the figures.

I believe that the high level of scrutiny in relation to these claims will begin with the understanding of the true protein level of CTLA-4 on the B-1a cells. As a consequence, this information should be presented in the most unambiguous way.

We thank the Reviewers for helpful critiques and constructive suggestions that have enabled us to improve our manuscript. Specific replies follow below.

Reviewer #1(Remarks to the Author):

The authors have responded to previous critique by providing a convincing and comprehensive analysis of mice lacking CTLA-4 specifically in B cells (or B-1a cells). The manuscript is clear, the data quality high and the findings should be of interest to a wide audience, given the importance of CTLA4 as a clinical target. While the mechanisms of CTLA4-mediated autoantibody production and spontaneous germinal center formation remain to be elucidated, the manuscript provides a comprehensive first description of CTLA4 as a regulator of B cell immunity. I have no further suggestions for improvement.

Reviewer #3 (Remarks to the Author):

As I stated in my previous report, given the exceptional groundbreaking nature of the authors' observations this manuscript will likely be very controversial and attract a very high level of scrutiny. I am still in disbelief that a very minor level of CTLA-4 on a small proportion of B-1a cells can have such dramatic functional impact in vivo. Although the authors made some efforts to suggest a putative underlying mechanism, it remains unconvincing how such a mechanism may operate. However, as I mentioned before, the functional data provided are convincing and conclusive, making the functional impact of CTLA-4 loss B-1a cells hard to dispute.

My only request, at this stage, is linked to the need to be completely open with the data. The authors' have been unable or unwilling to provide FACS data to document CTLA-4 expression on all B1-a populations used in the different experiments. I feel this should not be a difficult request.

We appreciate that the Reviewer considers our findings to be groundbreaking and that the functional data are convincing and conclusive. We understand Reviewer's concern and agree that the mechanism underlying CTLA-4 regulation of B-1a cells remains to be elucidated. Nevertheless, our studies to date demonstrate that CTLA4-4 is expressed by B-1a cells, negatively regulates their function. This regulation, as we show, is important for the maintenance of immune homeostasis. We also appreciate (and agree with) the Reviewer's preference for two-dimensional plots, and have provided these data (see below) in the current manuscript version.

1. A dot plot (not an histogram) should be included in the main figures to show the level of CTLA-4

expression on the cell populations displayed in Figure 1c. The authors provided the gating strategy but not the requested CTLA-4 dotplot. The dotplot is important to assess what are the characteristics of the cells that are marginally expressing more CTLA-4.

We agree with the Reviewer that the two-dimensional plot map is the preferable visualization of data. In the revised manuscript that we are submitting here, we have replaced the FACS histogram plots in *Figure 1c* with FACS contour plots, which quantitate the intracellular CTLA-4 expression in major mature B cell subsets and Treg cells from several mouse lines. As we show in this figure, splenic B-1a (sB-1a), peritoneal B-1a (pB-1a), FOB, MZB, peritoneal B-2 (pB-2) and splenic Treg (sTreg) cells from indicated mice are phenotypically defined and gated to show intracellular CTLA-4 expression. FACS plots: *Y-axis* shows surface CD5 expression for B cell subsets and intracellular Foxp3 expression for Treg cells; *X-axis* shows data for cells stained with anti-CTLA-4 antibody or isotype control antibodies conjugated with the same fluorochrome (PE) as the anti-CTLA-4 antibody.

The Reviewer may note that we use contour plots rather than dot plots. This is in keeping with the instructions to authors for the Nature Reporting Summary, which requires that all flow cytometry plots be contour plots or pseudocolor plots. We use contour plots in this study but could readily supply pseudocolor plots or any other format if the editors prefer.

2. As the authors use experimentally sB-1a cells from CTLA4^{+/+} (control), CTLA4^{fl/+} (heterozygous), and CTLA4^{fl/fl} (mutant), FACS dotplots (not histograms) showing the level of CTLA-4 on those three cell populations should be added to the figures.

We agree with Reviewer's point and have now added FACS contour plots, which now show the intracellular CTLA-4 expression in both splenic B-1a and peritoneal B-1a cells from CTLA-4 control, heterozygous and CKO mice in the revised manuscript. In this figure (*sFigure 2b*), splenic B-1a (sB-1a) and peritoneal B-1a (pB-1a) cells from indicated mice are defined and gated to show intracellular CTLA-4 expression. FACS plots: *Y-axis* shows surface CD5 expression and *X-axis* shows data for cells stained intracellularly with anti-CTLA-4 or isotype control antibody that is conjugated with the same fluorochrome (PE) as anti-CTLA-4 antibody.

Although the Reviewer only requested data for splenic B-1a cells, we have also added data for peritoneal B-1a cells in the figure. As we discussed in the Discussion section (*see page 17, second and third paragraph*), in both CKO and chimeric mice reconstituted with CTLA-4-deficient B-1a cells, the spleen is the principal lymphoid organ in which Tfh cells and GC responses occur. Nevertheless, we show that loss of CTLA-4 in CKO mice increases the frequency and total numbers of both splenic and peritoneal B-1a cells.

Further, although splenic B-1a express lower levels of CTLA-4 than peritoneal B-1a cells, both populations in CKO mice are dysfunctional in that they are able to induce Tfh cells and GC responses when transferred to and reconstituted in normal congenic recipients. Importantly, as our studies also shown, induction of Tfh cells and GC responses only proceeds when the recipients receive B-1a cells that do not express CTLA-4, as it does not happen when we transfer the peritoneal B-1a cells from CTLA4-heterozygous mice, despite that these cells well reconstitute B-1a cells in both spleen and peritoneal cavity of the recipients (*see page 12, second paragraph*).

I believe that the high level of scrutiny in relation to these claims will begin with the understanding of the true protein level of CTLA-4 on the B-1a cells. As a consequence, this information should be presented in the most unambiguous way.

We agree with the Reviewer. As now presented (corrected as requested by the Reviewer), we believe that our data are reproducible and scientifically valid as presented.

REVIEWERS' COMMENTS

Reviewer #3 (Remarks to the Author):

The author addressed all issues raised in my report. The manuscript is considerably improved. I suggest that the frequency of cells that are within the CTLA4+ in the new plots should be indicated (Fig1c and Supl Fig), for consistency with other FACS plots.

Reviewer #3 (Remarks to the Author):

The author addressed all issues raised in my report. The manuscript is considerably improved. I suggest that the frequency of cells that are within the CTLA-4⁺ in the new plots should be indicated (Fig1c and Supl Fig), for consistency with other FACS plots.

As the reviewer suggested, in Figure 1c and Supplementary Figure 2D of the revised manuscript, we have gated CTLA-4⁺ cells within the splenic B-1a and peritoneal B-1a cells and added the information of frequency of these cells.

We are thankful for the reviewer's helpful critiques and constructive suggestions that have enabled us to improve our manuscript. We also deeply appreciated his/her great efforts in mediating our responses to reviewer 4.